# How vulnerable is my policy? Adversarial attacks on modern behavior cloning policies

## Abstract

Learning from Demonstration (LfD) algorithms have shown promising results in robotic manipulation tasks, but their vulnerability to adversarial attacks remains underexplored. This paper presents a comprehensive study of adversarial attacks on both classic and recently proposed algorithms, including Behavior Cloning (BC), LSTM-GMM, Implicit Behavior Cloning (IBC), Diffusion Policy (DP), and VQ-Behavior Transformer (VQ-BET). We study the vulnerability of these methods to untargeted, targeted and universal adversarial perturbations. While explicit policies, such as BC, LSTM-GMM and VQ-BET can be attacked in the same manner as standard computer vision models, we find that attacks for implicit and denoising policy models are nuanced and require developing novel attack methods. Our experiments on several simulated robotic manipulation tasks reveal that most of the current methods are highly vulnerable to adversarial perturbations. We also show that these attacks are transferable across algorithms, architectures, and tasks, raising concerning security vulnerabilities to black-box attacks. In addition, we test the efficacy of randomized smoothing, a widely used adversarial defense technique, and highlight its limitation in defending against attacks on complex and multi-modal action distribution common in complex control tasks. In summary, our findings highlight the vulnerabilities of modern BC algorithms, paving way for future work in addressing such limitations.

## 1 Introduction

Learning from Demonstration (LfD) has emerged as a powerful paradigm in AI and robotics, enabling agents to acquire complex behaviors from expert demonstrations. These techniques are increasingly deployed in real-world scenarios, such as to enable robots in industrial automation and household robotics. However, these policies pose potential security risks since they can be easily maliciously manipulated by adversaries, causing undesired behaviors or even catastrophic incidents. Motivated by these risks, to the best of our knowledge, we are the first to present a systematic study of the vulnerabilities of LfD algorithms to adversarial attacks with a focus on modern behavior cloning algorithms.

Adversarial attacks are a widely studied area that aims to develop imperceptible perturbations (Szegedy et al., 2013) to change the output of machine learning models. Early work by Szegedy et al. (2013) and Goodfellow et al. (2014) revealed that adding small, imperceptible perturbations to images could drastically alter the prediction of neural network classifiers. Since then, a significant number of works have explored various attack methods and defense techniques against adversarial attacks (Akhtar & Mian, 2018; Zhang et al., 2020b; Chakraborty et al., 2021).

However, the robustness of LfD models to adversarial attacks has been largely overlooked in prior research, particularly in the context of robotic manipulation tasks. While previous works have examined adversarial robustness in reinforcement learning (Mo et al., 2023; Sun et al., 2020; Pattanaik et al., 2017; Lin et al., 2017; Gleave et al., 2019; Zhang et al., 2020a; 2021; Moos et al., 2022), including whitebox attacks (Huang et al., 2017; Casper et al., 2022) and backdoor detection (Chen et al., 2023), the impact of such perturbations on LfD remains underexplored, especially as they relate to modern BC methods. Attacks in the LfD domain present unique challenges, LfD policies must process temporal sequences of observations, handle

multimodal action distributions, and maintain stable control even under perturbations. This paper aims to investigate the vulnerabilities specific to LfD algorithms under adversarial perturbations, shedding light on their susceptibility and resilience in robotic settings.

Our work focuses specifically on post-deployment white-box attacks, where an adversary has access to the trained model parameters but cannot modify the training process, as well as black-box transfer attacks. Unlike training-time attacks that aim to corrupt the learning process, our attacks target the inference phase, attempting to cause task failures through carefully crafted perturbations to visual observations. While this may seem like a strong adversarial capability, the increasing trend toward open-source release of robotic policy weights makes this a practical concern that needs to be addressed.

In this paper, we evaluate the adversarial robustness of several leading LfD frameworks, including Vanilla Behavior Cloning (BC), LSTM-GMM (Mandlekar et al., 2021), Implicit Behavior Cloning (IBC) (Florence et al., 2021), Diffusion Policy (Chi et al., 2023), and VectorQuantizied-Behavior Transformer (VQ-BET) (Lee et al., 2024). Among these, IBC and Diffusion Policy have unique design pipelines due to the fact that they are implicit rather than explicit policies and require more nuanced attacks. IBC employs energy-based models to learn implicit policies, offering greater flexibility in learning complex, multimodal behaviors compared to traditional methods. In IBC, the correct action is selected iteratively based on energy distribution rather than explicitly during inference. To address this, we introduce a sampling-based attack method that approximates the local energy surface, increasing the likelihood of selecting the desired target action. Diffusion Policy, on the other hand, uses a generative model approach with denoising diffusion techniques to iteratively refine actions, allowing it to capture diverse and continuous action distributions. While existing attacks (Chen et al., 2024) can degrade performance, they require manipulating the entire denoising process, leading to high compute costs of attack. One of our insights is that attacks are most effective at later stages of the denoising process. By applying attacks specifically only on later stages we can significantly improve attack efficiency. Overall, we hope our results serve as an impetus for enhancing awareness of the security and reliability concerns regarding policies learned via behavior cloning and will inspire researchers to develop more robust LfD algorithms.

The primary contributions of our work are as follows:

- We conduct the first comprehensive study of white-box adversarial attacks, both online (Projected Gradient Descent (PGD) (Madry et al., 2017)) and offline (Universal Adversarial Perturbation (Moosavi-Dezfooli et al., 2016)) attacks, on a variety of different behavior cloning algorithms. We are the first to study adversarial attacks on LSTM-GMM, IBC, VQ-BET algorithms. We evaluate these attacks in both targeted and untargeted settings and highlight the vulnerability of all the LfD algorithms studied in this paper.

- We propose novel attack formulations for implicit models such as IBC and Diffusion Policy. Our work addresses the unique challenges posed by the iterative action selection process, and propose efficient attacks that only require manipulating a few steps of the denoising process.

- We provide insights into the transferability of attacks across algorithms, tasks, and vision backbones. We find surprising transferability of the attacks across all the variations, highlighting the importance of considering robustness in the training pipeline of modern imitation learning.

- We also show the limited efficacy of the commonly used adversarial defense, Randomized Smoothing (Cohen et al., 2019), and discuss the implications of using such defenses in real-time and multi-modal distributional settings such as robotics.

- We find that, out of all the policies we test, Diffusion Policies are the most robust. While recent work (Carlini et al., 2023) has shown that combining a pretrained denoising diffusion probabilistic model and a standard high-accuracy classifier can yield robustness for image classifiers, we are the first to study and showcase the relative robustness of diffusion policies. We provide evidence that this robustness might stem from its multi-step prediction process rather than inherent resilience. Our results show that reducing the prediction horizon significantly decreases the adversarial robustness of diffusion policies.

## 2 Related Work

There has been relatively little prior work exploring adversarial attacks on behavioral cloning algorithms. Hall et al. (2020) provide an early demonstration that vanilla behavioral cloning policies can be vulnerable to adversarial perturbations in driving simulations. Wu et al. (2023) also explore attacks against a vanilla BC policy trained on image observations. Boloor et al. (2019) focus on simple CNN-based BC and demonstrate the vulnerability of end-to-end autonomous driving models to simple physical adversarial examples such as black lines on the road. We note that all three papers only consider white-box attacks and only attack the steering angle while we attack high-dimensional robot control inputs and consider both white-box and black-box attacks.

Jia et al. (2022) show that adversarial patches can directly attack object detectors causing robots to grasp a human hand rather than a card. Unlike our work, which examines the vulnerabilities of modern LfD algorithms, they target traditional object detection models in industrial robots. Other orthogonal work to ours has examined adversarial attacks on simulated assistive robots trained via RL (He et al., 2023), has investigated demonstration set poisoning attacks to prevent downstream behavioral cloning (Zhan et al., 2020), and has studied language-based universal attacks (Zhao et al., 2024) for language-conditioned policies.

To the best of our knowledge, the prior work most related to this paper is by Chen et al. (2024), who explore adversarial attacks on diffusion policies. Their method involves attacking the entire denoising process in the diffusion policy, which is computationally expensive. By contrast, we demonstrate effective attacks by manipulating only a few steps of the denoising process, significantly reducing the attack cost (interms of time and compute). In addition, while Chen et al. only focus on developing attacks for a single type of policy learning algorithm, our research examines the vulnerabilities of several different LfD algorithms and also explores how adversarial perturbations transfer across different algorithms, tasks, and and visual backbones, offering a broader and more comprehensive perspective on the vulnerability of modern behavior cloning and the generalizability of such attacks.

In summary, while prior work such as (Hall et al., 2020), (Wu et al., 2023), and Boloor et al. (2019) demonstrate the vulnerability of individual CNN-based behavior cloning (BC) models in driving domains, our work is the first to systematically study adversarial robustness across a diverse suite of modern imitation learning architectures, including LSTM-GMM, VQ-Transformer, Implicit BC, and Diffusion Policy. Chen et al. (2024), who focus exclusively on diffusion models, we introduce tailored attack methods for a broad range of architectures and evaluate them side-by-side. Finally, our work is the first to study black-box attacks via transferability evaluations across algorithms, tasks, and vision backbones, offering insights into shared vulnerabilities that are critical for the design of robust IL systems. Together, we believe these contributions highlight the novelty and importance of our work.

## 3 Behavior Cloning Algorithms

In this section, we provide a brief background and high-level overview of the Behavior Cloning (BC) algorithms we study in this paper. To provide clarity throughout the discussion, we first define some key notations used across these algorithms. Let $\xi \in \Xi$ represent a set of expert trajectory demonstrations, where $\xi$ is a trajectory consisting of a sequence of state-action pairs $(s, a)$, sampled from an expert policy $\pi^*(s)$. Our objective in behavior cloning is to learn a policy $\pi_\theta(s)$, parameterized by $\theta$, that imitates the expert's behavior by minimizing a loss function $L$ that measures the difference between the expert actions and the actions predicted by the learned policy.

Formally, for a given policy $\pi_\theta$, we aim to minimize: $\pi_\theta^* = \arg\min_{\pi_\theta} \sum_{\xi \in \Xi} \sum_{s \in \xi} L(\pi_\theta(s), \pi^*(s))$, where $L$ is typically the cross-entropy loss for discrete actions or mean squared error for continuous actions.

We study five different behavior cloning approaches in this work. Due to space and because these are well-known algorithms we provide only a high-level description here. For more details, please see Appendix B.:

**Vanilla Behavior Cloning (BC)** learns a direct mapping from states to actions through supervised learning. Given state-action pairs $(s, a)$, it trains a neural network to minimize the mean squared error

(continuous actions). While effective for simple tasks, BC struggles with multimodal behaviors and suffers from compounding errors during execution.

**LSTM-GMM** enhances the performance of BC by incorporating temporal dependencies through an LSTM network and modeling multimodal action distributions using Gaussian Mixture Models (GMMs). At each timestep $t$, the LSTM processes the state sequence and outputs GMM parameters, enabling the policy to capture both temporal dynamics and complex action distributions. The policy $\pi_\theta(a_t|s_t, h_{t-1})$ is trained by maximizing the likelihood of expert actions under the predicted GMM distributions.

**Implicit Behavior Cloning (IBC)** reformulates policy learning using energy-based models. Instead of directly predicting actions, IBC learns an energy function $E_\theta(s, a)$ that assigns low energy to expert actions and high energy to other actions. The policy is implicitly defined as $\pi_\theta(s) = \arg\min_a E_\theta(s, a)$ and is trained using contrastive learning to minimize the energy of expert actions relative to sampled negative actions.

**Diffusion Policy (DP)** takes a generative approach by learning to iteratively denoise actions. Starting from Gaussian noise $a_T$, the policy refines actions through $T$ denoising steps conditioned on the state: $a_{t-1} = \alpha(a_t - \gamma\varepsilon_\theta(s, a_t, t)) + \sigma\mathcal{N}(0, I)$, where $\varepsilon_\theta$ is a learned noise prediction network. This approach enables modeling of complex, continuous action distributions.

**VQ-Behavior Transformer (VQ-BET)** combines transformer architectures with vector quantized VAEs to handle multimodal continuous actions. The policy discretizes the action space using a hierarchical codebook and predicts actions by minimizing L1 loss. This approach balances expressiveness and tractability while capturing both coarse and fine-grained action details.

## 4 Adversarial Attacks on Imitation Learning

### 4.1 Adversarial Attack Methods

In our study, we focus on two primary adversarial attack methods that have shown significant effectiveness in computer vision tasks:

**Projected Gradient Descent (PGD)** (Madry et al., 2017) is an iterative attack method that generates adversarial perturbations by taking multiple steps in the direction that maximizes the loss function. Starting from an initial input $x_0$, PGD iteratively updates the input using the following rule: $x_{k+1} = \Pi_{B_\epsilon(x_0)}(x_k + \alpha \cdot \text{sign}(\nabla_x J(\theta, x_k, y)))$, where $J(\theta, x_k, y)$ is the model's loss function, $\alpha$ is the step size, and $\Pi_{B_\epsilon(x_0)}$ denotes projection onto the $\ell_p$-norm ball of radius $\epsilon$ centered at $x_0$. The projection step ensures that the perturbed input remains within the allowable perturbation bound. PGD is considered a strong attack as it explores the local loss landscape thoroughly through multiple iterations.

**Universal Adversarial Perturbations (UAP)** (Moosavi-Dezfooli et al., 2016) aims to find a single perturbation vector that can cause misclassification when applied to multiple different inputs. Unlike PGD which computes perturbations for each input independently, UAP generates a single perturbation $v$ that satisfies $\|v\|_p \le \epsilon$ and successfully affects a high fraction of the input distribution. The perturbation is computed iteratively over a set of training inputs by accumulating the minimal perturbation needed to cause misclassification for each input while maintaining the constraint on the perturbation magnitude. This approach is particularly relevant for real-world scenarios where computing per-input perturbations may not be feasible, and the same attack vector needs to work across multiple observations.

**Targeted vs Untargeted Attacks:** Targeted and untargeted attacks represent two distinct adversarial objectives. Untargeted attacks aim to degrade the policy's performance by maximizing the loss function, causing the model to output any incorrect action that leads to task failure. They focus on disrupting normal policy execution without specifying a particular alternative behavior. In contrast, targeted attacks attempt to force the policy to output specific undesired actions by minimizing the distance between the policy's output and a chosen target action. For example, in a robotic manipulation task, an untargeted attack might cause the robot to move erratically or freeze in place, while a targeted attack would intentionally guide the robot's motion toward a specific undesired location or trajectory. This distinction is important because

targeted attacks often require more sophisticated optimization strategies to achieve precise control over the policy's behavior.

## 4.2 Threat Model

Before describing our adapted attacks, we first clearly specify our threat model:

This threat model is particularly relevant for deployed robotic systems using large pre-trained policies, where model weights are publicly available but the training process is complete. We study both online attacks (PGD) that can adapt perturbations in real-time and offline attacks (UAP) that must generate a single fixed perturbation designed to work across all states.

**Adversary's Goal**: The attacker aims to cause task failure by perturbing visual observations during deployment, either through untargeted perturbations that disrupt normal policy execution or targeted perturbations that force specific undesired actions.

**Adversary's Knowledge and Capabilities**: The attacker has white-box access to the trained policy parameters but cannot modify them. Perturbations are limited to the visual observation space (no direct action manipulation). Perturbations must remain within an $L_p$ norm ball of radius $\epsilon$ to maintain imperceptibility. The attacker can compute gradients through the entire policy network. We note that eef pose, eef quaterions, and gripper state as input along with the camera images; however, our attacks are only applied to the vision component of the state.

**Transferability Attacks**: We would like to emphasize that we also study black-box transfer attacks. In particular, we study how adversarial perturbations can transfer across different models, architectures and tasks without requiring access to the target model's parameters or training data. In these kinds of transfer settings, an attacker can generate effective perturbations using only a surrogate model with similar architecture, making these attacks feasible in black-box scenarios where the target model's internals are inaccessible. This property significantly broadens the threat surface, as successful attacks can be mounted with minimal knowledge of the target system.

## 4.3 Attacks on Explicit BC Algorithms

The objectives for running PGD, and UAP on explicit behavior cloning methods such as Vanilla BC, LSTM-GMM, and VQ-BET are based directly on their respective loss functions. For Vanilla BC, the adversarial attacks aim to maximize the mean squared error (MSE) loss between the predicted and expert actions by introducing small perturbations to the input states. In LSTM-GMM, the attacks target the temporal dependencies modeled by the LSTM and the multimodal action distributions captured by the Gaussian Mixture Model (GMM), aiming to disrupt the likelihood maximization over the GMM outputs. For VQ-BET, the attacks exploit the latent action space by targeting the prediction loss of discrete latent codes, ultimately leading to suboptimal action predictions. Each attack (PGD, UAP) thus aims to create adversarial perturbations that exploit the specific vulnerabilities of these loss functions to degrade performance.

## 4.4 Attacks on Implicit BC Algorithms

Implicit BC algorithms differ from explicit ones in modeling the learning process and action selection, causing naive adversarial attacks largely fail to generate feasible perturbations. In this section, we propose new attack formulations for implicit BC models considering their unique designs.

### 4.4.1 Implicit Behavior Cloning

Implicit Behavior Cloning (IBC) leverages implicit modeling techniques and contrastive learning to learn a policy directly from expert demonstrations. In IBC, the policy $\pi_\theta(\mathbf{a} \mid \mathbf{s})$ is learned by optimizing an energy-based model (EBM) that assigns low energy values to actions demonstrated by the expert and higher energy values to other actions. The energy function $E_\theta(\mathbf{s}, \mathbf{a})$ parameterized by $\theta$ is trained using the InfoNCE loss, for a batch of $N$ actions:

$$\mathcal{L}_{\text{InfoNCE}} = \sum_{i=1}^{N} -\log \left( \frac{e^{-E_\theta(\mathbf{s}_i, \mathbf{a}_i)}}{e^{-E_\theta(\mathbf{s}_i, \mathbf{a}_i)} + \sum_{j=1}^{N_{\text{neg.}}} e^{-E_\theta\left(\mathbf{s}_i, \tilde{\mathbf{a}}_i^j\right)}} \right) \tag{1}$$

where $\tilde{\mathbf{a}}_i^j$ for $j = 1, \dots N_{\text{neg.}}$ are the negative samples and $i = 1 \dots N$ are the training samples in the batch. The parameters $\theta$ are optimized by minimizing $\mathcal{L}_{\text{InfoNCE}}$, encouraging the model to assign lower energy to expert actions compared to negative samples. This approach allows IBC to capture complex and multimodal action distributions, leading to more robust imitation of expert behaviors (Florence et al., 2021). Since IBC uses an implicit model with iterative sampling procedure for selecting actions, we need to develop specific formulations for untargeted and targeted attacks for these kinds of implicit models, specifically for the Derivative-Free Optimizer version of the inference. We summarize our attack in Algorithm 1.

For the targeted attack, a key challenge arises from the contrastive nature of IBC's loss function. Since the model can only minimize energy for actions present in the sample set, and these samples are drawn randomly, there is no straightforward way to directly optimize for a specific target action. The probability of the desired target action appearing in the negative samples is very low, making it difficult to construct an end-to-end loss function that reliably guides the model toward selecting this target action. Hence, we formulate the problem as finding the perturbation $\delta$ for a target action $\mathbf{a}'$ such that, $\tilde{p}_\theta(\mathbf{a}'|\mathbf{s} + \delta) > \tilde{p}_\theta(\mathbf{a}|\mathbf{s} + \delta)$. To achieve this, *we introduce a sampling-based attack method to approximate the local energy surface*, making it easier to select the target action. Specifically, we randomly sample a small number of negative actions, along with the target action, to estimate the energy surface around the target region. We also consider the original action during the attack. we then iteratively perform gradient ascent to decrease the energy of the target action compared to both the original and negative actions. However, in-order to further increase the probability of the target action being chosen during inference, we repeat this procedure $N_{iter}$ times to decrease the energy of the target action with respect to more actions that could possibly be selected due to their vicinity to the original actions. The details are presented in Algorithm 1.

For the untargeted attack, the objective is to perturb the state $s$ by finding a perturbation $\delta$ that increases the energy (reduces the probability) of the actions that would normally be taken by the trained policy on clean state observations. In this case, we aim to push the model towards selecting less optimal actions by maximizing the energy associated with the learned actions in the perturbed state. To achieve this, we perform gradient ascent on the input pixels to maximize the energy of the correct action ($a_{clean}$ in Algorithm 1). Thus, by maximizing the energy of the correct action, the policy is forced to select alternative actions that are perturbed away from the intended action along different dimensions of the action space. While these selected actions maintain some structure from the original state-action distribution, the perturbations cause sufficient deviation to disrupt successful task execution.

### 4.4.2 Diffusion Policy

Diffusion Policy (DP) (Chi et al., 2023) aims to overcome the necessity of approximating the normalizing constant (the negative samples required in the above IBC method) in an energy based model, by learning the score function of the action-distribution.

In particular, the score function is defined as the gradient of the log-conditional probability distribution of actions, which is usually learnt as a noise-prediction network ($\varepsilon_\theta$) parameterized by $\theta$.

$$\nabla_{\mathbf{a}} \log p(\mathbf{a} \mid \mathbf{s}) = -\nabla_{\mathbf{a}} E_\theta(\mathbf{s}, \mathbf{a}) \approx -\varepsilon_\theta(\mathbf{s}, \mathbf{a}) \tag{2}$$

Starting from $\mathbf{a}^k$ sampled from Gaussian noise, DP iteratively denoises the sample $k$ times to get a desired noise-free sample $\mathbf{a}^0$.

$$\mathbf{a}^{k-1} = \alpha \left( \mathbf{a}^k - \gamma \varepsilon_\theta \left( \mathbf{s}, \mathbf{a}^k, k \right) + \mathcal{N} \left( 0, \sigma^2 I \right) \right) \tag{3}$$

where $\alpha, \gamma, \sigma$ are the hyper-parameters that collectively define the noise schedule. The complete inference is defined in Appendix B.

Algorithm 2 shows our online attack method for Diffusion Policy. Both targeted and untargeted attacks use Mean Squared Error (MSE) loss for propogation of gradient. For the targeted attack, we try to minimize the

---

**Algorithm 1** Implicit BC PGD Attack

---

**Require:** Trained energy model $E_\theta(s, a)$, state $s$, observation $o$, number of samples $N_{samples}$, number of iterations $N_{iters}$, decay rate $K$, perturbation bound $\epsilon$, step size $\alpha$
1: Obtain clean action $a_{clean}$ by running IBC on $s$
2: **for** $epoch = 1, 2, \ldots, N_{epochs}$ **do**        ▷ Optimize over multiple samples
3:     Initialize sample set $\mathcal{S} = \{\tilde{a}^i\}_{i=1}^{N_{samples}} \sim \mathcal{U}(a_{min}, a_{max})$
4:     **if** Targeted **then**
5:       Introduce $a_{clean}, a_{target}$ into $\mathcal{S}$
6:     **end if**
7:     **for** $iter = 1, 2, \ldots, N_{iters}$ **do**        ▷ Inner PGD attack iterations
8:       $\{E_i\}_{i=1}^{|\mathcal{S}|} \leftarrow \{E_\theta(s', \tilde{a}^i)\}_{i=1}^{|\mathcal{S}|}$        ▷ Compute energies with perturbation
9:       $\{\tilde{p}_i\}_{i=1}^{|\mathcal{S}|} \leftarrow \left\{ \frac{e^{-E_i}}{\sum_{j=1}^{|\mathcal{S}|} e^{-E_j}} \right\}_{i=1}^{|\mathcal{S}|}$        ▷ Compute softmax probabilities
10:      **if** Targeted **then**
11:        Compute cross-entropy loss with $a_{target}$ as the true label:
12:        Loss $= -\log(\tilde{p}_{\text{target}})$        ▷ $\tilde{p}_{\text{target}}$ is the probability of $a_{target}$
13:      **else**
14:        Compute untargeted loss:
15:        Loss $= -E_\theta(s', a_{clean})$        ▷ Maximize energy of the correct action for untargeted attacks
16:      **end if**
17:      Update $s'$ using PGD step:
18:      $s' = s' + \alpha \cdot (\nabla_{o_t} \text{Loss})_{\mathcal{B}_\epsilon}$        ▷ Projected on the $l_p$ norm ball
19:     **end for**
20: **end for**
21: **return** $s'$

---

**Algorithm 2** Diffusion Policy PGD Attack

---

**Require:** Observation horizon $T_0$, Action Horizon $T_a$, Prediction Horizon $T_p$, State sequence $\mathbf{S}_t = \{\mathbf{s}_{t-T_o+1}, \ldots, \mathbf{s}_t\}$, number of denoising iterations $K$
**Ensure:** Action sequence $\mathbf{A}_t = \{\mathbf{a}_t, \ldots, \mathbf{a}_{t+T_p-1}\}$
1: $\mathbf{A}_t^{clean} = $ Diffusion Policy Inference$(\mathbf{S}_t)$
2: $\mathbf{A}_t^{target} = \mathbf{A}_t^{clean} + $ Desired Perturbations        ▷ Only for Targeted Attack
3: Initialize $T_{\text{attack}}, \epsilon, \alpha, \gamma, \sigma, N_{iters}$
4: Initialize $\mathbf{A}_t^{(K)} \sim \mathcal{N}(\mathbf{0}, \mathbf{I})$
5: **for** $k = K, K-1, \ldots, 1$ **do**
6:     $\mathbf{A}_t^{(k-1)} = \alpha(\mathbf{A}_t^{(k)} - \gamma \epsilon_\theta(\mathbf{S}_t, \mathbf{A}_t^{(k)}, k)) + \sigma \mathcal{N}(\mathbf{0}, \mathbf{I})$
7:     **if** $k < T_{\text{attack}}$ **then**        ▷ Attack during the last $K - T_{attack}$ timesteps
8:      **for** $N_{iters}$ **do**        ▷ Inner PGD iterations
9:       **if** Targeted **then**
10:        $Loss = -\text{MSELoss}(\mathbf{A}_t^{k-1}, \mathbf{A}_t^{target(k-1)})$
11:       **else**
12:        $Loss = \text{MSELoss}(\mathbf{A}_t^{k-1}, \mathbf{A}_t^{clean(k-1)})$
13:       **end if**
14:       $\mathbf{S}_t = \mathbf{S}_t + \alpha \cdot \nabla_{\mathbf{O}_t}(Loss)_{\mathcal{B}_\epsilon}$        ▷ Grad. ascent w.r.t current observations and project on $\epsilon$ ball.
15:      **end for**
16:     **end if**
17: **end for**
18: **return** $\mathbf{S}_t$

---

distance between our predicted action and the target action (line 10) by doing gradient descent. Whereas, in the untargeted attack, we try to maximize the distance between the predicted action and the clean action (line 12) by doing gradient ascent. Running this attack end-to-end during the whole denoising process can be costly, as we need to backpropogate through the entire network for each iteration for a single inference step. However, we can reduce this computation by taking inspiration from prior work on image editing attacks (Salman et al., 2023) and only apply the perturbations during last timesteps (line 7 of Algorithm 2) of the denoising process. This enables us to avoid wasting attacks when the actions are very random (during the initial steps of denoising) and only apply the attack when the data has started to converge towards the mode. This significantly reduces the attack effort when compard with prior work (Chen et al., 2024) (in our case by 80%, since we use $T_{attack} = 80$ and $K = 100$) while not affecting the quality of adversarial attack.

# 5 Experiments & Results

We design our experiments to the answer to following questions: (1) How vulnerable are modern behavior cloning algorithms to adversarial attacks? (2) How transferable are the attacks across different algorithms and different tasks? (3) What is the impact of different feature extraction backbones on attack performance, as in how transferable are the attacks between different vision architectures? (4) How does the action prediction horizon of the diffusion policy affect its vulnerability? (5) How do adversarial defense techniques perform in our setting and what are the takeaways? (6) How sensitive are attacks to the range of adversarial perturbation?

## 5.1 Environments

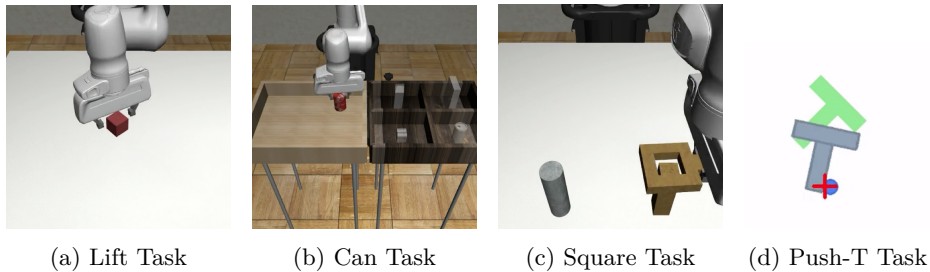

(a) Lift Task     (b) Can Task     (c) Square Task     (d) Push-T Task

Figure 1: Environments used to study adversarial robustness of modern behavior cloning algorithms. (a)-(c) are from RoboMimic (Mandlekar et al., 2021) and (d) is from Florence et al. (2021)

To demonstrate the adversarial robustness of modern behavior cloning algorithms, we consider a diverse set of common benchmarks shown in Figure 1. The tasks of Lift, Can and Square are taken from Robomimic (Mandlekar et al., 2021), where the state-of-the art frameworks such as Diffusion Policy and LSTM-GMM have been shown to have a nearly 100% success rate in non-adversarial settings. To further assess the ability of adversarial attacks to breach these frameworks on a diverse task, we consider the Push-T environment, first introduced by Florence et al. (2021) and then subsequently used by Diffusion Policy (Chi et al., 2023) and VQ-BET (Lee et al., 2024). **Lift** is a foundational manipulation task where a robot arm must lift a small cube from a table surface. **Can** is a manipulation task requiring the robot to transfer a soda can from a large source bin into a smaller target bin. **Square** is a high-precision manipulation task where the robot must pick up a square nut and insert it onto a vertical rod. **Push-T** is a contact-rich manipulation task where the robot must guide a T-shaped block to a fixed target location using a circular end-effector. Further descriptions and details of these tasks are included in Appendix A.

## 5.2 Pretrained Policies

To provide consistent and reproducible results, we attack the pre-trained checkpoints for LSTM-GMM, IBC and Diffusion Policy released by the authors of Diffusion Policy (Chi et al., 2023) on these suite of tasks, and train our policies for Vanilla-BC and VQ-BET, due to absence of publicly available checkpoints. We evaluate all the environments on 50 randomly initialized environments across 3 different seeds for reporting the mean and standard deviation of the success rate.

## 5.3 How vulnerable are modern behavior cloning algorithms to adversarial attacks?

To assess the vulnerability of modern behavior cloning algorithms to adversarial attacks, we conducted a comprehensive evaluation using both online (PGD) and offline (UAP) attack methods. Answering question (1) our findings, as illustrated in Figures 2 and 3, reveal significant vulnerabilities in the adversarial robustness of current algorithms when faced with perturbations in the observation space. Among the algorithms tested, VQ-BET demonstrated the highest susceptibility to adversarial perturbations. We hypothesize that this vulnerability stems from the discrete nature of its action space, which may lead to discontinuous decision boundaries. In contrast, algorithms employing iterative methods for action selection, such as IBC and

Diffusion Policy, exhibited relatively higher robustness. This enhanced resilience can be attributed to the inherent stochasticity in their action selection processes during inference. It is important to note that the effectiveness of these attacks varies depending on the complexity of the task environment. For instance, the Lift environment allows for a larger margin of error, making it more forgiving to substantial perturbations in actions. However, as task complexity increases, we observe a dramatic reduction in the robot task success rates (increase in attack success rates) across all algorithms. For example, Mandlekar et al. (2021) categorize the difficulty of the tasks with Lift being the easiest, Can being harder than Lift, and Square being harder than Can. As we increase the complexity of the task, we notice an increase in the efficacy of the adversarial attacks as detailed in Appendix D.

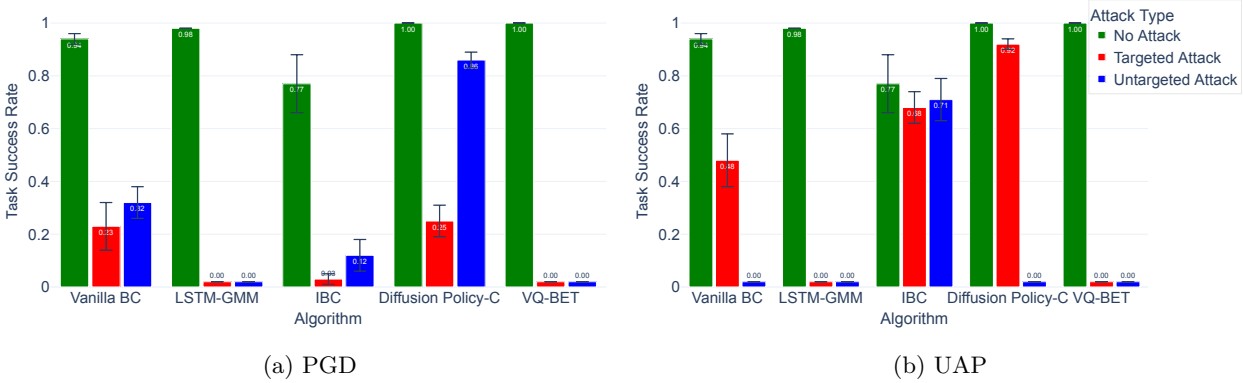

Figure 2: Comparison of PGD and UAP attacks for the Lift task. The y-axis denotes the average success rates of different BC policies, where lower is better for the attacker.

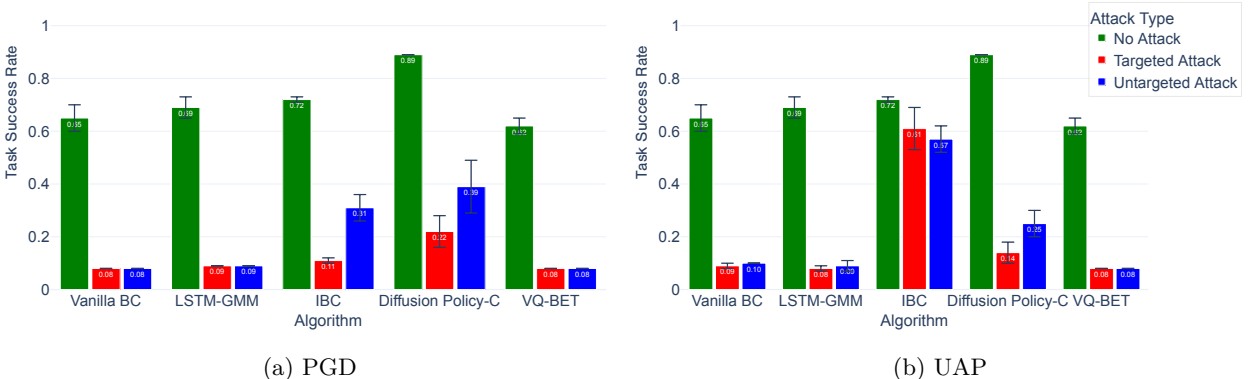

Figure 3: Comparison of PGD and UAP attacks for the Push-T task. The y-axis denotes the average success rates of different BC policies, where lower is better for the attacker.

Particularly concerning is our finding that these vulnerabilities persist even with minimal perturbations - even when epsilon is restricted to values as small as 4/256 (approximately 1.5% of the input range), we observe substantial degradation in performance across most algorithms. This heightened sensitivity to small perturbations suggests a fundamental brittleness in current behavior cloning approaches, as even well-constrained adversarial attacks can significantly compromise policy performance (Fig. 12 in Appendix G).

Table 1: Average Success rates of different target policies under untargeted universal adversarial perturbations (UAP) generated from various attacker policies in the Lift task. The rows correspond to the attacker policies (random refers to Gaussian noise with mean zero and std epsilon), and the columns correspond to the target policies. Each cell reports the absolute task success rate after applying the attack.

| Attacker \ Target Policy | Vanilla BC | LSTM-GMM | IBC | DiffusionPolicy-C | VQ-BET |
|---|---|---|---|---|---|
| Random | 0.96 | 0.84 | 0.80 | 1.00 | 0.98 |
| Vanilla BC | 0.00 | 0.00 | 0.80 | 1.00 | 0.94 |
| LSTM-GMM | 0.94 | 0.00 | 0.72 | 1.00 | 0.96 |
| IBC | 1.00 | 0.10 | 0.64 | 1.00 | 0.98 |
| DiffusionPolicy-C | 0.82 | 0.22 | 0.78 | 0.00 | 0.94 |
| VQ-BET | 0.94 | 0.50 | 0.84 | 1.00 | 0.00 |

Table 2: Average success rates of target policies on the Push-T task under untargeted UAPs generated from different attacker policies. This table illustrates the cross-algorithm transferability of adversarial perturbations.

| Attacker \ Target Policy | Vanilla BC | LSTM-GMM | IBC | DiffusionPolicy-C | VQ-BET |
|---|---|---|---|---|---|
| Random | 0.60 | 0.61 | 0.64 | 0.82 | 0.55 |
| Vanilla BC | 0.10 | 0.08 | 0.41 | 0.80 | 0.22 |
| LSTM-GMM | 0.15 | 0.09 | 0.33 | 0.78 | 0.31 |
| IBC | 0.14 | 0.08 | 0.14 | 0.71 | 0.17 |
| DiffusionPolicy-C | 0.27 | 0.10 | 0.24 | 0.14 | 0.22 |
| VQ-BET | 0.26 | 0.14 | 0.47 | 0.61 | 0.08 |

Table 3: Percentage decrease in robot task completion rate (i.e., success rate reduction) when adversarial perturbations generated on the Lift task are applied to other tasks (Can, Square). Negative values indicate counterintuitive increases in performance under attack. This table highlights multi-task transferability of universal perturbations.

| Algorithm \ Task | LIFT | CAN | LIFT-TO-CAN | SQUARE | LIFT-TO-SQUARE |
|---|---|---|---|---|---|
| Vanilla BC | 100% | 100% | 50% | 100% | -40% |
| LSTM-GMM | 100% | 100% | 40% | 100% | 38.9% |
| IBC[*] | 7.9% | 100% | 100% | 100% | 100% |
| DiffusionPolicy-C | 100% | 100% | 45% | 100% | 6.6% |
| VQ-BET | 100% | 100% | 0% | 100% | 11.4% |

[*]IBC has very low (almost zero) performance on the Can and Square task, so the above metric may not capture the full picture for (only) IBC.

## 5.4 Can adversarial examples transfer across different algorithms and tasks?

The transferability of adversarial examples across different behavior cloning algorithms presents an intriguing phenomenon, given the substantial differences in their loss functions and training methodologies (as detailed in Section 3). While these algorithms share a common image encoder (ResNet-18), their end-to-end training approach results in distinct feature representations that are not easily interpretable.

In simpler environments like the Lift task (see Table 1), where baseline success rates are high (>90% for most algorithms), we observed limited transferability with relatively small proportional drops in performance, aligning with our initial expectations. Intriguingly, in the PushT environment (Table 2) we noticed high-transferability of attacks between algorithms except for Diffusion Policy and also as we progressed to more complex environments (e.g., Square in Table 8 in the Appendix), where baseline success rates are lower

and tasks are naturally less robust to action perturbations, we noticed that transferred attacks often caused larger proportional drops in performance relative to the baseline.

To answer the question regarding whether the attacks developed for one environment can transfer to other environments keeping the algorithm fixed, we evaluated the transferability of the attacks across environments. Our results in Table 3, where we report the percentage decrease in the robot task completion rate compared to the non-attacked version, show that the attacks developed in Lift can transfer to both the other environments (Can and Square). However in rare case of BC for Square, we see an unexpected increase in performance when attacked using the attack developed for Lift environment but this could be due to random initialization of environments and the time constraint for testing only 3 seeds for each algorithm. It could also be due to the fact that adding a small amount of action noise to policies can sometimes increase performance by helping the policy get unstuck.

## 5.5 Robustness of Implicit Policies

While our results show that both IBC and Diffusion Policy are generally more robust than explicit policies, the asymmetry in their robustness to different attack types warrants deeper analysis. Specifically, Diffusion Policy demonstrates greater resilience to PGD attacks, which we hypothesize stems from its multi-step denoising inference process. Since PGD introduces perturbations that are propagated through a long sequence of refinement steps, the model's iterative structure may naturally attenuate their impact—effectively smoothing out sharp adversarial gradients over time. Conversely, IBC exhibits stronger robustness to universal perturbations, likely due to its stochastic, sampling-based action selection. Because IBC selects actions based on contrastive energy scores over a diverse set of samples, a fixed UAP is less likely to consistently degrade performance across all sampled actions. Moreover, while PGD directly optimizes the energy surface in IBC, aligning with the model's objective and making it a stronger threat, UAP lacks this fine-grained adaptation. These differences suggest that each model's inference dynamics play a central role in shaping its vulnerability, and future work could explore this further via step-wise attack tracing or ablations over the inference pipeline.

## 5.6 What is the impact of different feature extraction backbones to attack performance?

We further wanted to gain insights about the transferability of attack with different vision-backbones used for policy learning. In-order to test this, we developed perturbations using ResNet-18 as the backbone and then deployed these attacks on policies that were trained using ResNet-50, without regenerating the attacks. This cross-architecture transfer scenario yielded surprising results. In the Lift task (see Table 4), we observed high transferability for some algorithms (e.g., LSTM-GMM and IBC), while others showed more resilience (e.g., Diffusion Policy-C and VQ-BET). The more complex Push-T task (see Table 5) demonstrated a more consistent pattern of partial transferability across all algorithms.

Table 4: Average Success rates before and after untargeted UAP attacks generated using a ResNet-18 backbone and evaluated on policies using either ResNet-18 or ResNet-50. This setup tests cross-architecture transferability of adversarial attacks on the Lift task. NA indicates no attack.

| Algorithm | NA Resnet-18 | NA Resnet-50 | Resnet-18 | Resnet-50 |
|---|---|---|---|---|
| Vanilla BC | 1.00 | 1.00 | 0.21 | 0.75 |
| LSTM-GMM | 1.00 | 1.00 | 0.00 | 0.25 |
| IBC | 0.95 | 0.50 | 0.85 | 0.38 |
| DiffusionPolicy-C | 1.00 | 1.00 | 0.00 | 1.00 |
| VQ-BET | 1.00 | 1.00 | 0.00 | 0.98 |

Notably, in many cases, the ResNet-50 models showed vulnerability to attacks developed for ResNet-18, suggesting that simply increasing model capacity does not guarantee improved robustness against cross-architecture attacks. It also highlights the existence of shared vulnerabilities across different network architectures, which adversarial perturbations can exploit even when transferred to a different backbone. These

Table 5: Average Success rates on the Push-T task under untargeted UAPs created using ResNet-18 and tested on ResNet-50-based policies. Demonstrates robustness of different algorithms to adversarial perturbations transferred across visual backbones. NA indicates no attack.

| Algorithm | NA Resnet-18 | NA Resnet-50 | Resnet-18 | Resnet-50 |
|---|---|---|---|---|
| Vanilla BC | 0.72 | 0.62 | 0.09 | 0.20 |
| LSTM-GMM | 0.72 | 0.56 | 0.08 | 0.21 |
| IBC | 0.74 | 0.57 | 0.63 | 0.27 |
| DiffusionPolicy-C | 0.88 | 0.78 | 0.14 | 0.54 |
| VQ-BET | 0.62 | 0.65 | 0.08 | 0.29 |

results underscore the importance of considering cross-architecture vulnerabilities in the design of robust behavior cloning systems.

### 5.7   How does the action prediction horizon of the diffusion policy affect its vulnerability?

In addition to the above experiments, we find an interesting trade-off between the action horizon of the Diffusion Policy and robustness. In Fig. 4, we observe that as the action horizon increases (the number of actions taken at a time), while keeping the prediction horizon the same, the policy shows increasing robustness to universal attack. We hypothesize that as the action horizon increases the number of times the perturbed observation gets observed decreases thus allowing for smaller compounding errors during the inference.

However, if the action horizon is too long then the latency and recovering from sub-optimal trajectories might lead to worse overall performance.

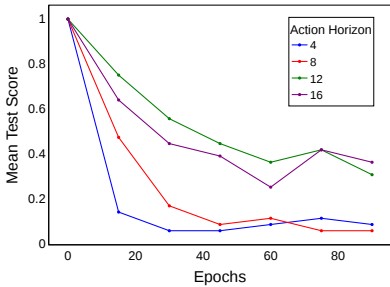

Figure 4: Test mean score vs epochs for action prediction horizon of 16 for lift, and various action horizons during Untargeted UAP.

### 5.8   How do adversarial defense techniques perform?

To evaluate the applicability of adversarial defenses from computer vision to robotic control, we investigate Randomized Smoothing (Cohen et al., 2019) (we provide further details about the defense technique and hyperparameters in Appendix E), a widely adopted defense technique, against our strongest attack method (PGD). Our experimental results, presented in Tables 6 and 7, reveal that the effectiveness of randomized smoothing varies significantly across different algorithms and task domains.

In the Lift task, we observe a spectrum of improvements in robustness. Most notably, Diffusion Policy and VQ-BET demonstrate remarkable enhancement in performance, with the task success rate with randomized smoothing increasing from 25% to 98% and 8% to 66% respectively. Vanilla BC exhibits moderate improvements, with success rates increasing from 48% to 52%, respectively. IBC also shows substantial gains, improving from 21% to 50% success rate, while LSTM-GMM demonstrates minimal benefits. The Push-T task, however, presents a markedly different scenario. The benefits of randomized smoothing are considerably attenuated across most algorithms, with IBC being the slight exception, showing improvement from 38% to 50% success rate. The limited effectiveness in this task may be attributed to the inherent multi-modal nature of the action distribution in the Push-T environment (Lee et al., 2024). In such cases, averaging predictions can lead to actions that lie between valid modes of the distribution, resulting in suboptimal behavior.

This discrepancy in effectiveness across tasks suggests that the utility of randomized smoothing is highly task-dependent. The defense appears more effective in tasks with simpler, unimodal action distributions, while its benefits diminish in scenarios involving complex, multi-modal action spaces. These findings highlight the need for defense mechanisms specifically designed for robotic control tasks that can maintain effectiveness across varying degrees of task complexity and action space characteristics.

Table 6: Average Success rates of various behavior cloning algorithms on the Lift task under different adversarial and defense conditions. We compare the baseline performance (no attack), randomized smoothing alone, PGD attack, and PGD combined with randomized smoothing to assess the effectiveness of smoothing-based defenses.

| Algorithm | NA | NA Randomized Smoothing | PGD Attack | Randomized Smoothing with PGD |
|---|---|---|---|---|
| Vanilla BC | 1.00 | 1.00 | 0.48 | 0.52 |
| LSTM-GMM | 1.00 | 0.93 | 0.00 | 0.00 |
| IBC | 0.95 | 0.80 | 0.21 | 0.50 |
| DiffusionPolicy-C | 1.00 | 1.00 | 0.25 | 0.98 |
| VQ-BET | 1.00 | 1.00 | 0.08 | 0.66 |

Table 7: Average Success rates of behavior cloning algorithms on the Push-T task under baseline, PGD, randomized smoothing, and combined PGD + smoothing settings. This table highlights the varying impact of defense strategies in a contact-rich manipulation environment.

| Algorithm | NA | NA Randomized Smoothing | PGD Attack | Randomized Smoothing with PGD |
|---|---|---|---|---|
| Vanilla BC | 0.74 | 0.74 | 0.08 | 0.08 |
| LSTM-GMM | 0.66 | 0.54 | 0.00 | 0.00 |
| IBC | 0.68 | 0.67 | 0.38 | 0.50 |
| DiffusionPolicy-C | 0.88 | 0.84 | 0.23 | 0.24 |
| VQ-BET | 0.72 | 0.71 | 0.10 | 0.10 |

## 5.9 How sensitive are attacks to the range of adversarial perturbation?

Our analysis reveals vulnerabilities in behavior cloning algorithms even with minimal perturbations (for Universal Untargeted Attacks). As shown in Figure 12, while decreasing epsilon values generally reduces attack efficacy, algorithms like VQ-BET, LSTM-GMM, and Diffusion Policy still exhibit substantial performance degradation even at very small epsilon values ($\epsilon$ of 4/256). This heightened sensitivity to small perturbations highlights a concerning vulnerability in current behavior cloning approaches, suggesting that even well-constrained adversarial attacks can significantly compromise policy performance.

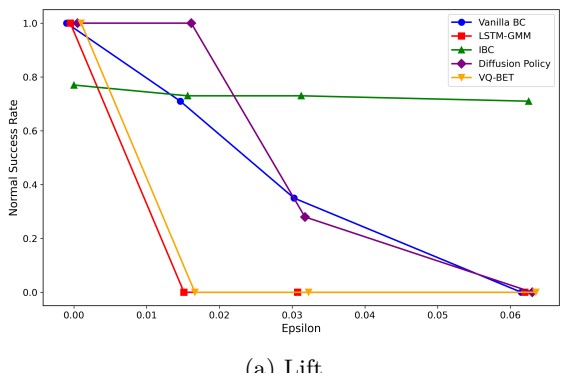
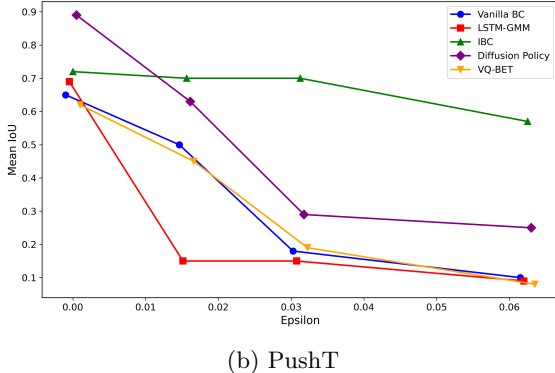

(a) Lift

(b) PushT

Figure 5: Task success rates of behavior cloning algorithms under decreasing perturbation magnitudes ($\epsilon$), demonstrating their sensitivity to even small adversarial inputs. The steep drop in performance emphasizes the lack of robustness across algorithms.

# 6   Conclusion & Future Work

Our results show that modern behavior cloning algorithms are vulnerable to adversarial attacks. Interestingly, implicit policies such as Implicit Behavior Cloning and Diffusion Policy seem to be more robust than the explicit policies. However, our results also demonstrate that the attack success rate is dependent on the task. As tasks gets harder, it becomes easier to attack these algorithms. This also holds true based on the results from transferability of attacks between different algorithms. Our results provide evidence that the different algorithms as well as the same algorithm trained with a different architecture are learning some similar features that are not completely orthogonal. Thus posing a security challenge since even if we are using different vision encoders, task, or policy these perturbations are transferable.

We believe that our work lays foundation for future work in the direction of adversarial robustness of LfD policies. Future work includes designing better metrics to capture the nuanced effects of adversarial attacks on trajectories, rather than relying solely on success rates. Such metrics could provide deeper insights into the uncertainty in the state-action distributions learned by the policies. Furthermore, while a much progress on adversarial robustness has been made in computer vision, the sequential nature of robotic LfD policies and the complex relationship between vision representations and resulting actions (especially for implicit policies such as IBC and Diffusion Policy) provides an underexplored and exciting are for future research into both vulnerabilities and defenses. In-addition, adversarial attacks with respect to the physical parameters can provide reliability estimates and better coverage of state-action distribution. Our current results show that even in settings where the robot feedback is more than vision, adversarial attacks only on vision are significant enough to defeat a policy. Future work should investigate other types of hybrid statespaces such as domains with force sensing feedback along with visual feedback to see if visual attacks alone are sufficient to significantly degrade policy performance. Finally, future work should investigate physical attacks on real robot platforms to study whether our findings in simulation generalize to real world deployment of the different BC algorithms we have studied in this paper.

# 7   Broader Impact

This work reveals significant vulnerabilities in behavior cloning algorithms that could affect the safe deployment of robotic systems in real-world applications. While exposing these security weaknesses could potentially enable malicious exploitation, we believe their disclosure is crucial for developing robust safety measures before widespread deployment in sensitive environments like industrial automation or healthcare robotics. Our systematic evaluation provides concrete guidance for developing more secure systems. Our goal is to ensure reliable and safe operation of robotic systems by identifying and addressing potential vulnerabilities before deployment, while encouraging the research community to prioritize adversarial robustness in future algorithm development.

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

## A   Task Description

We evaluate the vulnerability of behavior cloning methods on several manipulation tasks of varying complexity.

- **Lift**: A foundational manipulation task where a robot arm must lift a small cube (4cm x 4cm x 4cm) from a table surface. The task tests basic pick-and-place capabilities and serves as an entry-level benchmark. Success is determined by elevating the cube above a threshold height. Initial cube poses are randomized with z-axis rotation within a small square region at the table center.

- **Can**: A manipulation task requiring the robot to transfer a soda can from a large source bin into a smaller target bin. This task presents increased difficulty over Lift due to the more complex grasping requirements of the cylindrical can and the constrained placement target. The can's initial pose is randomized with z-axis rotation anywhere within the source bin.

- **Square**: A high-precision manipulation task where the robot must pick up a square nut and insert it onto a vertical rod. This task significantly increases complexity by requiring precise alignment and complex insertion dynamics. The nut's initial pose is randomized with z-axis rotation within a square region on the table surface.

- **Push-T:** A contact-rich manipulation task adapted from (Florence et al., 2021) where the robot must guide a T-shaped block to a fixed target location using a circular end-effector. The task requires precise control of contact dynamics, as the robot must strategically apply point contacts to maneuver the block along the desired trajectory. Unlike pick-and-place tasks, success depends on understanding and exploiting the complex dynamics of planar pushing. We evaluate using RGB image observations augmented with end-effector proprioception. Initial positions of both the T-shaped block and the end-effector are randomized to ensure learned policies must generalize across different pushing strategies.

## B   More Details on Behavior Cloning Algorithms

### B.1   Vanilla Behavior Cloning

Vanilla Behavior Cloning (Vanilla BC) learns a policy via supervised learning (Bain & Sammut, 1995; Torabi et al., 2018). Given a dataset of state-action pairs $\mathcal{D} = \{(s_t, a_t)\}_{t=1}^{N}$, it directly maps states to actions using a neural network, $\pi_\theta$ trained to minimize the mean squared error (MSE) for continuous actions:

$$\mathcal{L}_{\text{BC}}(\theta) = \frac{1}{N} \sum_{t=1}^{N} \|\pi_\theta(s_t) - a_t\|_2^2 \tag{4}$$

While effective for simple tasks, Vanilla BC struggles with tasks requiring long-term dependencies owing to the problem of compounding error and the tasks with multimodalilty in expert behavior, as it assumes a unimodal distribution over actions (Ross et al., 2011; Florence et al., 2021).

### B.2   LSTM-GMM

Long Short-Term Memory with Gaussian Mixture Model (LSTM-GMM) (Mandlekar et al., 2021) enhances Vanilla BC by incorporating temporal dependencies through an LSTM network (Hochreiter & Schmidhuber, 1997). The LSTM processes a sequence of states $s_1, s_2, \ldots, s_T$ recursively, maintaining an internal hidden state $h_t$ at each time step. The policy $\pi_\theta(a_t|s_t, h_{t-1})$ is parameterized by the LSTM to model the temporal structure, while a GMM captures multimodal action distributions at each time step. At each time step $t$, the LSTM updates its hidden state and predicts a multimodal distribution over actions : $h_t = \text{LSTM}(s_t, h_{t-1})$ and $p(a_t|s_t, h_{t-1}, \theta) = \text{GMM}(h_t)$. The policy is trained by maximizing the likelihood of the observed actions given the state sequence: $\pi_\theta = \arg\max_\theta \sum_{\xi \in \Xi} \sum_{t=1}^{T} \log p(a_t|s_t, h_{t-1}, \theta)$, where $p(a_t|s_t, h_{t-1}, \theta)$ is the

probability of action $a_t$ under the GMM, conditioned on the current state $s_t$ and the previous hidden state $h_{t-1}$.

$$h_t = \text{LSTM}(s_t, h_{t-1}) \tag{5}$$

$$p(a_t|s_t, h_{t-1}, \theta) = \text{GMM}(h_t) \tag{6}$$

$$\pi_\theta = \arg\max_\theta \sum_{\xi \in \Xi} \sum_{t=1}^{T} \log p(a_t|s_t, h_{t-1}, \theta) \tag{7}$$

## B.3 VQ-BET

The Vector Quantized Behavior Transformer (VQ-BET) (Lee et al., 2024) combines a transformer-based architecture with vector quantization to handle multi-modal continuous action spaces. The policy discretizes actions into latent codes using a hierarchical quantization process, which allows the model to capture both coarse- and fine-grained action details. The model's policy is formulated as a sequence prediction problem, where the transformer predicts discrete latent codes and continuous offsets for actions.

The Vector-Quantized Behavior Transformer (VQ-BeT) is a two-stage model for behavior cloning. It discretizes continuous action chunks using a Residual Vector-Quantized VAE (VQ-VAE) and trains a GPT-style transformer to predict the resulting latent action tokens from observation sequences.

**Action Tokenization via Residual VQ-VAE**   Given a continuous action chunk $a_{t:t+n}$, an encoder $\phi$ maps it to a latent vector:

$$x = \phi(a_{t:t+n})$$

This latent is discretized using a residual vector quantization process over $N_q$ codebooks:

$$z_q(x) = \sum_{i=1}^{N_q} z_q^i, \quad \text{where } z_q^i \in \mathcal{C}^i = \{e_1^i, \ldots, e_K^i\}$$

Each $z_q^i$ is selected via nearest-neighbor lookup from codebook $\mathcal{C}^i$.

A decoder $\psi$ reconstructs the original action:

$$\hat{a}_{t:t+n} = \psi(z_q(x))$$

The VQ-VAE is trained to minimize a combination of reconstruction and commitment losses:

$$\mathcal{L}_{\text{VQ}} = \|a_{t:t+n} - \psi(z_q(x))\|_1 + \|\text{sg}[x] - e\|_2^2 + \lambda_{\text{commit}}\|x - \text{sg}[e]\|_2^2$$

where $\text{sg}[\cdot]$ denotes the stop-gradient operator and $e$ is the selected embedding vector. This loss uses the standard VQ-VAE loss with stop-gradient tricks. The second term encourages the codebook vector $e$ to move toward the encoder output $\phi(a)$. The last term encourages the encoder output to stay close to the selected codebook vector. The coefficient $\lambda_{\text{commit}} = 1$ and controls strongly the encoder is "committed" to using codebook entries rather than drifting away. Each residual quantization layer has its own codebook, and the same VQ loss structure is applied layer-by-layer.

**Transformer-Based Code Prediction and Offset Correction**   A transformer $\pi_\theta$ is trained to predict the code indices from a sequence of past state observations $s_{t-h:t}$. Let $\zeta_{\text{code}}^i(s_t)$ denote the categorical logits output for codebook $i$, and let $c^i$ be the ground-truth code index (from quantization) for that codebook.

The predicted codes are decoded as:

$$\lfloor a_{t:t+n} \rfloor = \psi \left( \sum_{i=1}^{N_q} e_{c^i}^i \right)$$

An offset network $\zeta_{\text{offset}}$ adds a residual correction for improved action fidelity:

$$\tilde{a}_{t:t+n} = \lfloor a_{t:t+n} \rfloor + \zeta_{\text{offset}}(s_t)$$

**Code Prediction Loss**   To train the transformer's categorical heads, a focal loss is applied to each predicted codebook distribution:

$$\mathcal{L}_{\text{code}} = \mathcal{L}_{\text{focal}}(\zeta_{\text{code}}^1(s_t), c^1) + \beta \sum_{i=2}^{N_q} \mathcal{L}_{\text{focal}}(\zeta_{\text{code}}^i(s_t), c^i)$$

Here, $\beta$ is a weighting coefficient for secondary codebooks and where

$$\mathcal{L}_{\text{focal}} = -\alpha_t (1 - p_t)^\gamma \log(p_t). \tag{8}$$

**Total VQ-BeT Loss**   The total training loss combines code prediction and offset correction:

$$\mathcal{L}_{\text{VQ-BeT}} = \mathcal{L}_{\text{code}} + \|a_{t:t+n} - \tilde{a}_{t:t+n}\|_1$$

### B.4   Implicit Behavior Cloning

Implicit Behavior Cloning (IBC) (Florence et al., 2021) reformulates the problem of policy learning as an energy-based model (EBM). Instead of explicitly predicting actions, IBC defines a compatibility score between states and actions using an energy function $E_\theta(s, a)$. The policy is implicitly represented by selecting actions that minimize the energy: $\pi_\theta(s) = \arg\min_a E_\theta(s, a)$

$$\pi_\theta(s) = \arg\min_a E_\theta(s, a) \tag{9}$$

The model is trained using contrastive learning, where the energy of expert actions is minimized relative to negative (non-expert) samples. The training loss typically follows the InfoNCE objective, as discussed in more detail in section 4.4.1.

---

**Algorithm 3** Implicit BC Inference

---

**Require:** Trained energy model $E_\theta(s, a)$, observation $s$, number of samples $N_{samples}$, number of iterations $N_{iters}$, initial sampling std. dev. $\sigma_{init}$, decay rate $K$

1: Initialize $\{\tilde{a}^i\}_{i=1}^{N_{samples}} \sim \mathcal{U}(a_{min}, a_{max})$, $\sigma = \sigma_{init}$
2: **for** $iter = 1, 2, \ldots, N_{iters}$ **do**
3:     $\{E_i\}_{i=1}^{N_{samples}} \leftarrow \{E_\theta(s, \tilde{a}^i)\}_{i=1}^{N_{samples}}$         ▷ Compute energies
4:     $\{\tilde{p}_i\}_{i=1}^{N_{samples}} \leftarrow \left\{ \frac{e^{-E_i}}{\sum_{j=1}^{N_{samples}} e^{-E_j}} \right\}_{i=1}^{N_{samples}}$     ▷ Compute softmax probabilities
5:     **if** $iter < N_{iters}$ **then**
6:         $\{\tilde{a}^i\}_{i=1}^{N_{samples}} \leftarrow \text{Multinomial}(N_{samples}, \{\tilde{p}_i\}_{i=1}^{N_{samples}}, \{\tilde{a}^i\}_{i=1}^{N_{samples}})$   ▷ Resample with replacement
7:         $\{\tilde{a}^i\}_{i=1}^{N_{samples}} \leftarrow \{\tilde{a}^i + \mathcal{N}(0, \sigma)\}_{i=1}^{N_{samples}}$         ▷ Add noise
8:         $\{\tilde{a}^i\}_{i=1}^{N_{samples}} \leftarrow \text{clip}(\{\tilde{a}^i\}_{i=1}^{N_{samples}}, a_{min}, a_{max})$         ▷ Clip to bounds
9:         $\sigma \leftarrow K\sigma$         ▷ Shrink sampling scale
10:     **end if**
11: **end for**
12: $i = \arg\max_i(\{\tilde{p}_i\}_{i=1}^{N_{samples}})$
13: **return** $\tilde{a}^i$

---

### B.5   Diffusion Policy

Diffusion Policy (DP) (Chi et al., 2023) uses a novel generative approach to model action distributions by leveraging Denoising Diffusion Probabilistic Models (Ho et al., 2020). Diffusion Policy formulates visuomotor control as a conditional denoising diffusion process over action sequences. Let $\mathbf{a}_{1:T_p}^0$ denote a ground-truth action sequence of length $T_p$, and let $\mathbf{s}_{1:T_o}$ be a sequence of past observations (states), where $T_o$ is the observation horizon. The policy aims to model the conditional distribution $p_\theta(\mathbf{a}_{1:T_p} \mid \mathbf{s}_{1:T_o})$.

**Forward Process.** A forward noising process is defined by a Markov chain of Gaussian perturbations:

$$\mathbf{a}_{1:T_p}^k = \sqrt{\bar{\alpha}_k}\,\mathbf{a}_{1:T_p}^0 + \sqrt{1-\bar{\alpha}_k}\,\boldsymbol{\epsilon}, \quad \boldsymbol{\epsilon} \sim \mathcal{N}(0,I), \tag{10}$$

where $k \in \{1,\ldots,K\}$ is the diffusion step and $\bar{\alpha}_k$ is the cumulative product of a noise schedule.

**Policy Representation and Sampling.** The policy does not output actions directly but instead learns to denoise a noisy sample using a learned score function $\epsilon_\theta$. At inference time, a sample from the policy is produced by the reverse denoising process:

$$\mathbf{a}_{1:T_p}^{k-1} = \alpha_k\left(\mathbf{a}_{1:T_p}^k - \gamma_k\,\epsilon_\theta(\mathbf{s}_{1:T_o}, \mathbf{a}_{1:T_p}^k, k)\right) + \mathcal{N}(0, \sigma_k^2 I), \tag{11}$$

where $\alpha_k$, $\gamma_k$, and $\sigma_k$ are functions of the diffusion step $k$. This iterative procedure approximates sampling from the policy distribution $p_\theta(\mathbf{a}_{1:T_p} \mid \mathbf{s}_{1:T_o})$ via Langevin dynamics.

**Training Objective.** The training loss is derived from a variational bound on the negative log-likelihood:

$$\mathcal{L}_{\text{diff}} = \mathbb{E}_{\mathbf{s},\mathbf{a}^0,\boldsymbol{\epsilon},k}\left[\left\|\boldsymbol{\epsilon} - \epsilon_\theta(\mathbf{s}_{1:T_o}, \mathbf{a}_{1:T_p}^k, k)\right\|_2^2\right], \tag{12}$$

which corresponds to minimizing a weighted sum of denoising score-matching losses across diffusion steps. This loss indirectly maximizes a lower bound on the conditional log-likelihood:

$$\log p_\theta(\mathbf{a}_{1:T_p}^0 \mid \mathbf{s}_{1:T_o}) \geq \mathbb{E}_{q(\mathbf{a}_{1:T_p}^{1:K}\mid\mathbf{a}_{1:T_p}^0)}\left[\log p_\theta(\mathbf{a}_{1:T_p}^0 \mid \mathbf{a}_{1:T_p}^{1:K}, \mathbf{s}_{1:T_o})\right] - \text{KL}[q\|p], \tag{13}$$

where $q$ is the forward diffusion process and $p$ is the learned reverse process.

**Closed-loop Execution.** During deployment, the policy executes the first $T_a \leq T_p$ actions of the sampled trajectory and replans using updated state observations. This forms a receding-horizon loop that balances responsiveness and temporal consistency.

For Diffusion policy (Algorithm 4) we use absolute positional actions as the original work shows that CNN-based diffusion policy performs poorly with robomimic's offical dataset, that uses veloocity control, as in the actions are represented as delta with respect to the current.

---

**Algorithm 4** Diffusion Policy Inference

---

**Require:** Observation horizon $T_0$, Action Horizon $T_a$, Prediction Horizon $T_p$, State sequence $\mathbf{S}_t = \{\mathbf{s}_{t-T_o+1}, \ldots, \mathbf{s}_t\}$, number of denoising iterations $K$
**Ensure:** Action sequence $\mathbf{A}_t = \{\mathbf{a}_t, \ldots, \mathbf{a}_{t+T_p-1}\}$
1: Initialize $\alpha$, $\gamma$, $\sigma$
2: Initialize $\mathbf{A}_t^{(K)} \sim \mathcal{N}(\mathbf{0}, \mathbf{I})$
3: **for** $k = K, K-1, \ldots, 1$ **do**
4: $\quad \mathbf{A}_t^{(k-1)} = \alpha(\mathbf{A}_t^{(k)} - \gamma\epsilon_\theta(\mathbf{S}_t, \mathbf{A}_t^{(k)}, k)) + \sigma\mathcal{N}(\mathbf{0}, \mathbf{I})$
5: **end for**
6: **return** $\mathbf{S}_t$

---

## C  Hyperparameters

We adopt the following temporal horizons from Diffusion Policy:

- Action prediction horizon ($T_p$): 16 steps

- Action execution horizon ($T_a$): 8 steps

- Observation context window ($T_o$): 2 steps

For the adversarial attacks, we use the following settings:

1. Overall attack budget:
   - $\varepsilon = 0.0625$ (16/256) $L_\infty$ norm (normalized to input range $[0, 1]$)
   - Perturbations are clipped to $[0, 1]$ range

2. Framework-specific perturbation bounds:
   - For standard BC frameworks on Robomimic: $[0.15, 0.15, 0]$ in $(x, y, z)$ directions for relative end-effector positions
   - For Diffusion Policy on Robomimic: $[0.45, 0.45, 0]$ in $(x, y, z)$ directions for absolute end-effector positions. The larger perturbation magnitude accounts for the absolute position representation, compared to relative positions used in other frameworks
   - For all the frameworks on Push-T: $[100, 100]$ for the two action dimensions.

3. PGD attack parameters:
   - Number of iterations: 40
   - Per-iteration step size ($\varepsilon_{\text{iteration}}$): 0.005

For IBC inference, we use derivative-free optimization with $N_{\text{samples}} = 1024$.

## C.1   TARGET ACTION SELECTION

For targeted attacks across all algorithms, target actions are generated by perturbing the expected clean actions:

$$a_{target} = a_{clean} + \delta_{action} \tag{14}$$

where $a_{clean}$ is the action predicted by the unperturbed policy and $\delta_{action}$ is the desired action perturbation. For our experiments, we set $\delta_{action} = [0.15, 0.15]$ for perturbations in x and y directions for all frameworks except Diffusion Policy, where we use $\delta_{action} = [0.45, 0.45]$. These values were chosen to ensure the target actions remain within physically feasible bounds while being sufficiently different from the clean actions to potentially cause task failures. For PGD attacks, this target action computation is performed at each inference step using the current clean action prediction, while for UAP the target actions are computed once using the perturbed offline action trajectories.

# D   Results on Additional Environments

## D.1   Square Environment

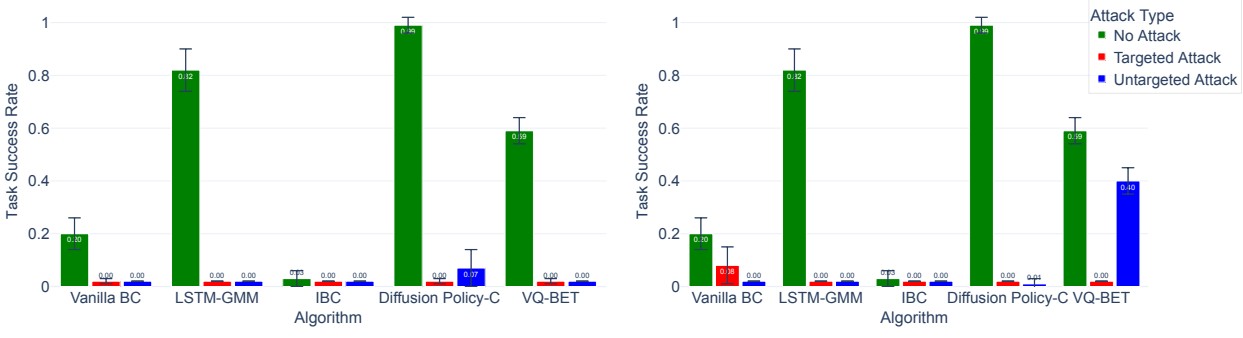

(a) PGD attacks for square.  (b) Universal perturbation attacks for square.

Figure 6: Comparison of PGD and universal perturbation attacks for square task.

Table 8: Inter-Algorithm Transferability of Universal Untargeted Perturbations for Square

| Attacker \ Target Policy | Square | | | | |
|---|---|---|---|---|---|
| | Vanilla BC | LSTM-GMM | IBC | DiffusionPolicy-C | VQ-BET |
| Random | 0.42 | 0.36 | 0.00 | 0.98 | 0.64 |
| Vanilla BC | 0.00 | 0.08 | 0.00 | 0.94 | 0.38 |
| LSTM-GMM | 0.18 | 0.00 | 0.00 | 0.96 | 0.62 |
| IBC | 0.42 | 0.1 | 0.00 | 0.98 | 0.62 |
| DiffusionPolicy-C | 0.00 | 0.00 | 0.00 | 0.00 | 0.32 |
| VQ-BET | 0.26 | 0.00 | 0.00 | 0.98 | 0.00 |

## D.2  Can Environment

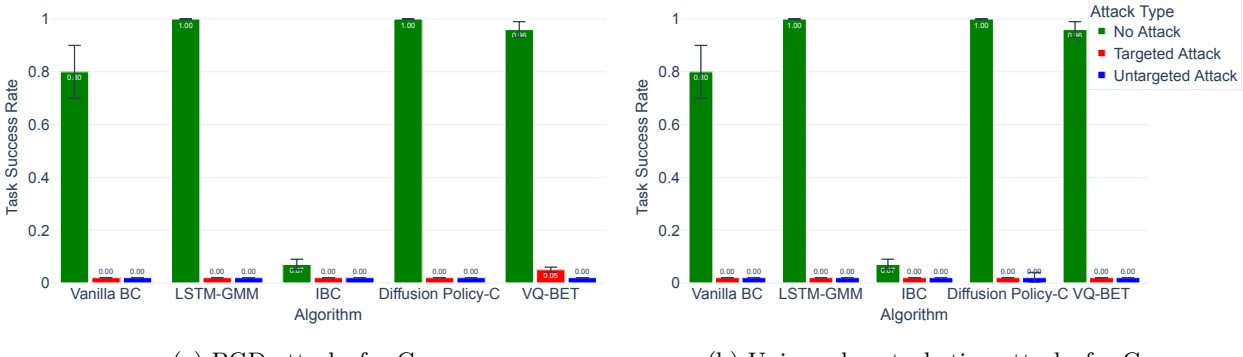

(a) PGD attacks for Can.  (b) Universal perturbation attacks for Can.

Figure 7: Comparison of PGD and universal perturbation attacks for Can task.

Table 9: Inter-Algorithm Transferability of Universal Untargeted Perturbations for Can

| Attacker \ Target Policy | Can | | | | |
| --- | --- | --- | --- | --- | --- |
| | Vanilla BC | LSTM-GMM | IBC | DiffusionPolicy-C | VQ-BET |
| Random | 0.62 | 0.94 | 0.00 | 1.00 | 0.96 |
| Vanilla BC | 0.00 | 0.66 | 0.00 | 0.42 | 0.88 |
| LSTM-GMM | 0.18 | 0.00 | 0.00 | 0.72 | 0.68 |
| IBC | 0.72 | 0.98 | 0.00 | 1.00 | 0.92 |
| DiffusionPolicy-C | 0.02 | 0.24 | 0.00 | 0.00 | 0.70 |
| VQ-BET | 0.34 | 0.64 | 0.00 | 0.42 | 0.04 |

Table 10: Inter-Architecture Transferability. Transferability of attacks trained with resnet-18 to resnet-50 as backbone for Can.

| Algorithm | NA Resnet-18 | NA Resnet-50 | Resnet-18 | Resnet-50 |
| --- | --- | --- | --- | --- |
| Vanilla BC | 0.75 | 0.70 | 0.00 | 0.34 |
| LSTM-GMM | 1.00 | 0.25 | 0.00 | 0.00 |
| IBC | 0.09 | 0.00 | 0.00 | 0.00 |
| DiffusionPolicy-C | 1.00 | 0.875 | 0.00 | 0.30 |
| VQ-BET | 1.00 | 0.70 | 0.04 | 0.70 |

## E    Randomized Smoothing

Randomized smoothing is a technique used to enhance the robustness of deep neural networks against adversarial perturbations. The core idea is to smooth the model's predictions by averaging over multiple randomly perturbed versions of the input. For a given input state $s$, the smoothed policy $\tilde{\pi}(s)$ is defined as:

$$\tilde{\pi}(s) = \mathbb{E}_{\varepsilon}[\pi(s + \varepsilon)], \quad \text{where} \quad \varepsilon \sim \mathcal{N}(0, \sigma^2 I) \tag{15}$$

During inference, we approximate this expectation by averaging predictions over $N$ randomly sampled perturbations:

$$\tilde{\pi}(s) \approx \frac{1}{N} \sum_{i=1}^{N} \pi(s + \varepsilon_i), \quad \text{where} \quad \varepsilon_i \sim \mathcal{N}(0, \sigma^2 I) \tag{16}$$

### E.1    Implementation Details

For our experiments, we used:

- Number of random samples ($N$): 100

- Noise standard deviation ($\sigma$):
  - Lift task: $\sigma = 0.1$
  - Push-T task: $\sigma = 0.05$

The $\sigma$ values were carefully chosen through validation to maintain performance on clean (non-attacked) inputs while providing meaningful defense against adversarial perturbations.

## F    ILLUSTRATIONS

In this section, we show examples of the adversarial perturbations. Figure 8 shows an example of *untargeted attacks* on the visual input for the Lift task. Figure 9 shows an example of *targeted attacks* on the visual

input for the Lift task. Figure 10 shows an example of *untargeted attacks* on the visual input for the Push-T task. Figure 11 shows an example of *targeted attacks* on the visual input for the Push-T task. We note that these perturbations are minor and in some cases almost imperceptible.

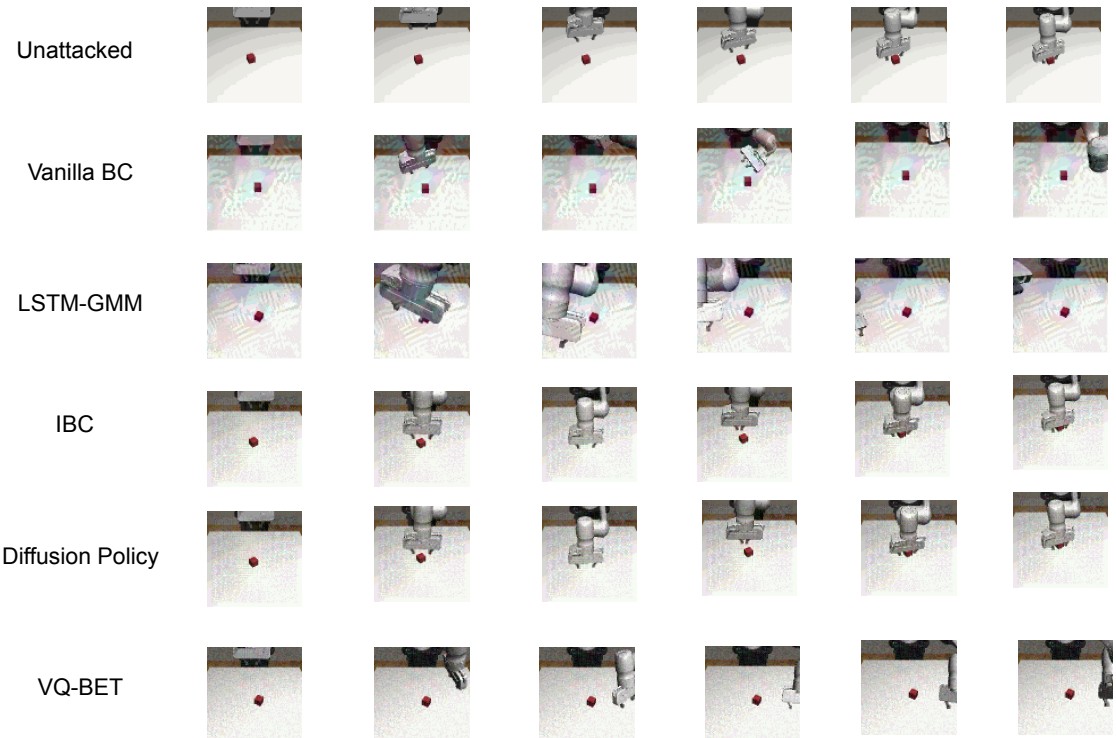

Figure 8: Untargeted Attacks on Lift task.

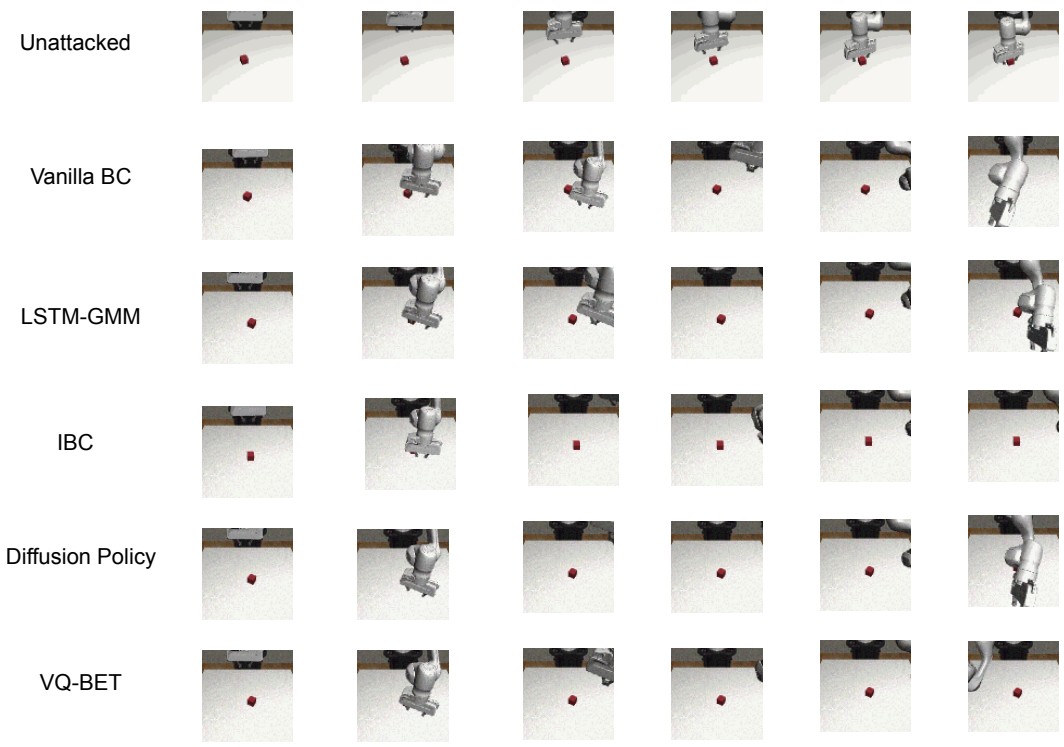

Figure 9: Targeted Attacks on Lift task, where the target direction is towards top-left corner of the object.

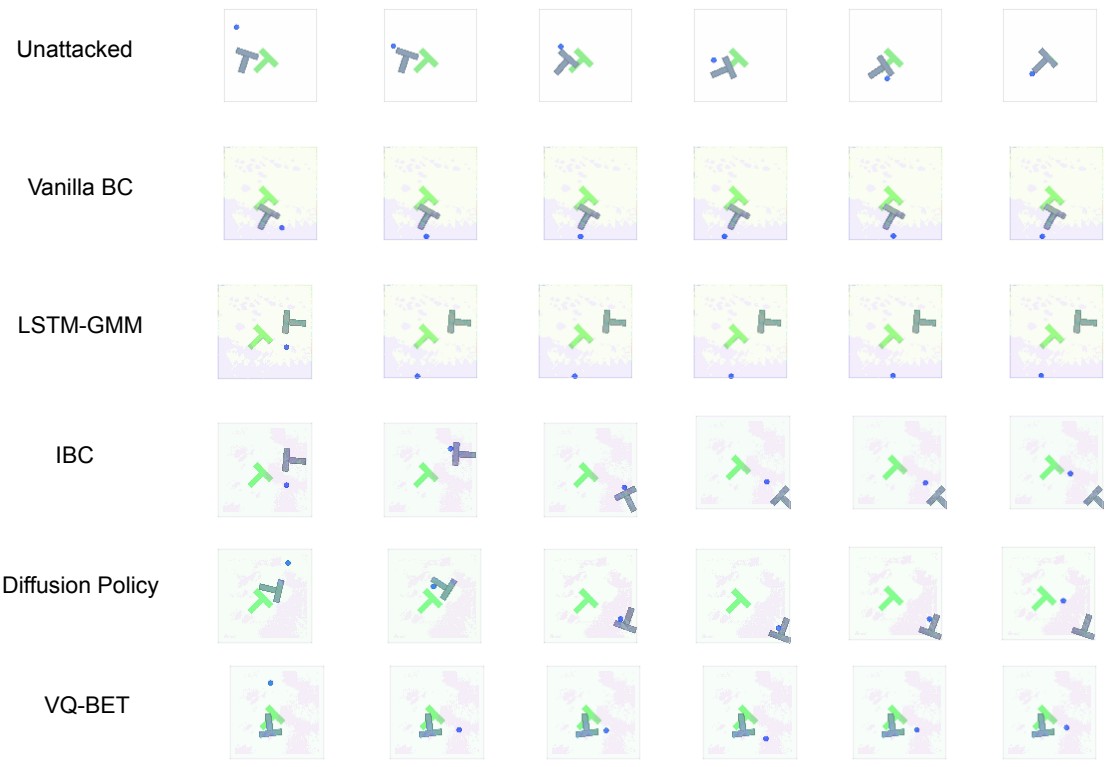

Figure 10: Untargeted Attacks on PushT task.

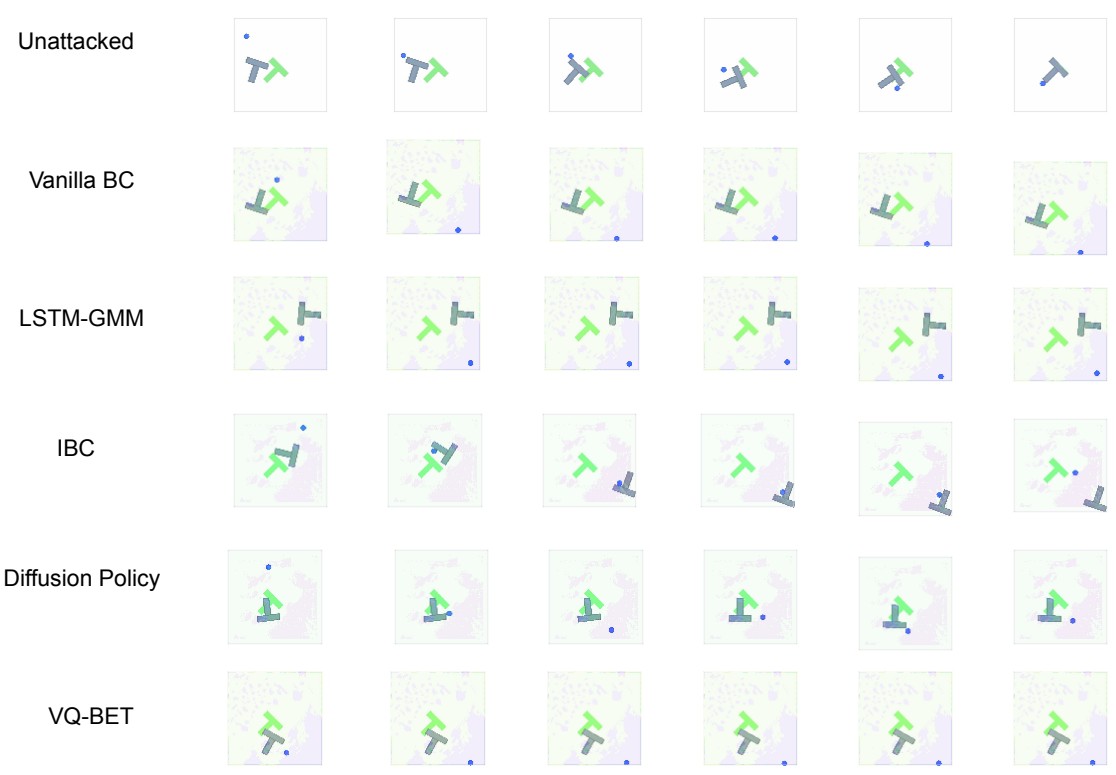

Figure 11: Targeted Attacks on PushT task, where the target is bottom right corner of the environment.

# G   Sensitivity to Epsilon Values

Our analysis reveals vulnerabilities in behavior cloning algorithms even with minimal perturbations (for Universal Untargeted Attacks). As shown in Figure 12, while decreasing epsilon values generally reduces attack efficacy, algorithms like VQ-BET, LSTM-GMM, and Diffusion Policy still exhibit substantial performance degradation even at very small epsilon values ($\epsilon$ of 4/256). This heightened sensitivity to small perturbations highlights a concerning vulnerability in current behavior cloning approaches, suggesting that even well-constrained adversarial attacks can significantly compromise policy performance.

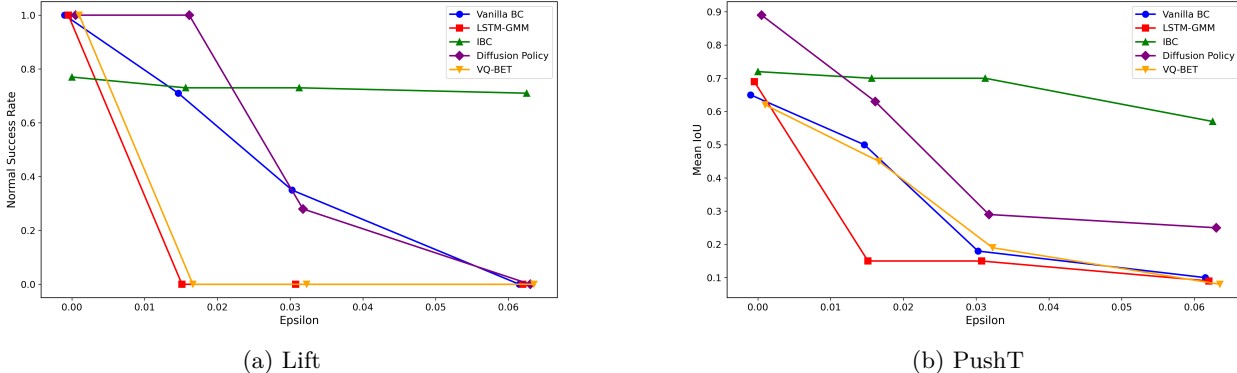

(a) Lift

(b) PushT

Figure 12: Performance of the algorithms to smaller epsilon values highlight the vulnerability and lack of robustness of the Behavior Cloning Algorithms.

