# OpenReview forum: "How vulnerable is my policy? Adversarial attacks on modern behavior cloning policies"
_TMLR — Rejected by TMLR_

### Review · Reviewer_X9sd · 2025-03-09

**Summary Of Contributions:**

This paper presents a systematic study of the vulnerability of modern Learning from Demonstration (LfD) algorithms to adversarial attacks. The experiment results somehow demonstrate the effectiveness of the proposed attack methods.

**Audience:**

Yes

**Claims And Evidence:**

No

**Requested Changes:**

1. Expand Beyond Chen’s Work: Ensure that the contributions clearly differentiate from prior work like Chen’s, and highlight novel elements that go beyond merely applying known methods to new tasks.

2. Enhance Experimental Complexity: Test the proposed methods on more complex and challenging tasks, such as dexterous manipulation or long-horizon tasks, to substantiate the empirical findings.

3. More importantly, please try to include real-world experiments: Conduct real-world experiments to address the gap between simulation and actual deployment, taking into account the impact of real-world factors on adversarial robustness.

**Strengths And Weaknesses:**

### Strengths:

1. Comprehensive Analysis: The paper provides a broad and thorough investigation of the adversarial vulnerabilities in modern LfD algorithms across multiple attack settings (targeted, untargeted) and models (BC, LSTM-GMM, IBC, DP, VQ-BET).

2. Different types of  Attack Strategies: It introduces different attack methods tailored for implicit models such as IBC and DP, showcasing how these models can be targeted despite their more complex design.

3. Cross-Algorithm Transferability: The study of attack transferability across various algorithms and tasks provides valuable insights into the generalizability of adversarial vulnerabilities, which is highly relevant for real-world applications.

---

### Weaknesses:

1. Lack of Novelty: The methods presented are essentially adaptations of image adversarial attack techniques to robotics, which lacks significant innovation. The approach doesn't provide any new concepts or mechanisms specifically designed for robotic systems.

2. Limited Contribution Compared to Prior Work: Compared to Chen's work, the contributions in this paper are minimal. Chen's paper also explores online/offline, targeted/untargeted attacks. Although this paper claims a broader experimental scope, particularly beyond Diffusion Policy, it doesn’t provide a groundbreaking advancement—more of a report on existing methods applied to different models.

3. Simplistic Experimental Setup: The experiments are conducted on extremely simple tasks like RoboMimic and Push-T, which are short-horizon and relatively easy. For a paper that aims to be more empirical, it would benefit from the inclusion of more challenging tasks such as dexterous manipulation or long-horizon tasks. Without these, the conclusions drawn from the experiments are hard to substantiate.

4. No Real-World Deployment: As a robotics paper, a major limitation is the lack of real-world experiments. The gap between simulation and real-world deployment is significant, and various factors such as delays and processing time in real-world systems are much more important than deploying on simulation platforms.

---

> ### Author Response · Authors · 2025-04-24
>
> > Lack of Novelty: The methods presented are essentially adaptations of image adversarial attack techniques to robotics, which lacks significant innovation. The approach doesn't provide any new concepts or mechanisms specifically designed for robotic systems.
>
> We respectfully disagree regarding the assessment of novelty. Much of ML research involves taking techniques from one domain and adapting them to a new domain or combining them in new ways and we believe that our work offers significant value to the TMLR research community. Furthermore, we note that we are the first to study both white-box adversarial attacks across a range of popular BC algorithms. We are also the first to study transferability attacks (black-box) across these algorithms. We argue that image adversarial attacks are a natural way and underexplored way to attack BC policies. Even though the attacks have existed in other settings, we argue that doing a thorough study of their effectiveness across different imitation learning tasks and algorithms is important for understanding the vulnerabilities of modern BC algorithms. While some of our techniques may not be especially novel, we believe that our experiments and results are quite novel. We strongly believe that our work is a good fit for TMLR which encourages submissions with interesting and valuable contributions, claims that are supported by evidence, and research of interest to the community, without putting large emphasis on subjective claims of novelty.
>
>
> > Ensure that the contributions clearly differentiate from prior work like Chen’s, and highlight novel elements that go beyond merely applying known methods to new tasks.
>
> Thank you for the suggestion to reinforce our contributions. As we acknowledge in our paper, Chen et al. (2024) performed the first investigation into adversarial attacks on diffusion-based policies. Our work significantly extends beyond this prior work by providing a comprehensive evaluation of adversarial vulnerabilities across a diverse set of modern imitation learning architectures, including LSTM-GMM, Implicit BC, and VQ-Transformer. We introduce various attack strategies—untargeted, targeted, and universal—and analyze their effectiveness and transferability across different models. We also show that we can optimize the attacks on Diffusion Policy leading to much cheaper attacks than those proposed by Chen et al. Furthermore, while Chen et al. only study white-box attacks, we also study black-box transfer attacks. Our analysis of transfer attacks across algorithms, tasks, and vision backbones has never before been studied in the context of BC algorithms.
>
> This broader scope and unified evaluation framework offer deeper insights into the adversarial robustness of contemporary IL policies, beyond the specific case of diffusion-based models addressed by Chen et al.. We have added more clarification regarding the contributions of our paper in a new dedicated related work section that discusses Chen et al. as well as other earlier work studying adversarial attacks on vanilla BC algorithms. We agree that this is important to ensure our contributions are clear.
>
> > Enhance Experimental Complexity: Test the proposed methods on more complex and challenging tasks, such as dexterous manipulation or long-horizon tasks, to substantiate the empirical findings.
>
> Thank you for your suggestion. We are in the process of adding a longer horizon image kitchen task based on the Franka Kitchen task from D4RL. We use the Franka Kitchen robotic manipulation environment introduced in (Gupta et al., 2019) with a Franka Panda arm with a 7 dimensional action space. This environment is a long-horizon task that requires the agent to complete a sequence of four tasks in order. While the original environment is state-based, we use an image-based variant of it by rendering the states with the MuJoCo renderer as a 112 by 112 image. We currently have results for VQ-BET showing that PGD attacks on VQBET drops our success rate (expected number of tasks in the 4 task sequence that are completed) from 2.86 to 0. Given this promising result on a more complex image-based adaptation of a D4RL benchmark we plan to add results for BC, LSTM-GMM, IBC, and Diffusion Policy.

---

> > ### Author Response · Authors · 2025-04-24
> >
> > > Please try to include real-world experiments
> >
> > We appreciate this suggestion and agree that an exciting area of future work is to explore real-world environments. Unfortunately, these experiments require acquiring expensive robot hardware, extensive experimental setup, hyperparameter tuning and testing, and are out of scope for our current paper and given our current resources. We also would like to note that the majority of prior work on adversarial robustness in sequential decision making, including the most recent work by Chen et al. (2024), only consider attacks in simulation. We believe there is strong value in both simulation studies and real-world case studies. We believe simulation is well suited for trying out many algorithms and attacks and that this type of analysis is common and well received in the ML community. We agree that real-world experiments are a great area for future research and have noted this in our Conclusion and Future Work section.

---

### Review · Reviewer_t4r7 · 2025-03-10

**Summary Of Contributions:**

This paper investigates the influence of white-box adversarial attacks (PGD and UAP) on different behavioral cloning algorithms, including naive BC, LSTM-GMM, IBC, DP, and VQ-BET. For implicit behavioral cloning algorithms, this paper introduces a new method which makes the standard attack fit. The authors further show the transferability of the learned attack across different tasks, suggesting the promise of attack learning without much white-box knowledge of the policy.

**Audience:**

Yes

**Claims And Evidence:**

No

**Requested Changes:**

- As mentioned in weakness 1, a clear formulation / definition of attack learning objective is necessary.
- Add experiments of applying robust RL attack instead of supervised learning attack, which makes the paper much more reasonable.
- Add more experiments results from other domains, e.g., D4RL benchmark.
- Since the title is "How vulnerable is my policy" and the key objective of this pape is to study the vulnerability of the BC policies, I believe the authors should put the fig.11 (the sensitivity to adversarial range $\epsilon$) to main text instead of appendix, which provides the key information for how vulnerable the policy is.

**Strengths And Weaknesses:**

**strengths**
- This paper studies the vulnerability of BC policies to adversarial attack, which is of the interest of related community (e.g., RL, robotics).
- The authors provides experiments results on extensive BC baselines.

**weaknesses**
- The adversarial attack is not clearly formulated. Although the authors give text explanations in sec.3.3~3.4, a mathematical equation / definition for each individual BC algorithm would be more clear.
- While this paper studies **the vulnerability of a policy** in a reinforcement learning task, a sequential decision making problem, the authors only perform PGD and UAV attacks, which is designed for supervised learning tasks. On the contrary, many well-known state-based attack methods in robust reinforcement learning are missing, e.g., [1] [2]. The key difference between attackers in RL and supervised learning is that the latter only perturbs the input to the **one-step** worst case, while the former perturbs to minimize the **total return**.
- Meanwhile, the authors only consider the one-step adversarial defence (e.g., randomized smoothing), which is designed for supervised learning. Other defence methods in robust RL [3] are absent.
- All experiments are conducted on Robomimic and push-T tasks. The tasks in robomimic has very similar styles while the push-T task is really simple. More experiments from diverse domains with different styles (e.g., mujoco / adroit / kitchen tasks in D4RL benchmarks) are needed, especially when the authors want to draw a generic conclusion on BC algorithms.

[1] Zhang, Huan, et al. "Robust deep reinforcement learning against adversarial perturbations on state observations." Advances in Neural Information Processing Systems 33 (2020): 21024-21037.

[2] Zhang, Huan, et al. "Robust reinforcement learning on state observations with learned optimal adversary." arXiv preprint arXiv:2101.08452 (2021).

[3] Moos, Janosch, et al. "Robust reinforcement learning: A review of foundations and recent advances." Machine Learning and Knowledge Extraction 4.1 (2022): 276-315.

---

> ### Author Response · Authors · 2025-04-24
>
> Thank you for your comments and helpful suggestions and questions! We agree that our transferability experiments show the promise of attack learning without direct white-box knowledge of the policy.
>
>
> > The adversarial attack is not clearly formulated. Although the authors give text explanations in sec.3.3~3.4, a mathematical equation / definition for each individual BC algorithm would be more clear.
>
> Thank you for this suggestion. We have added additional details in Appendix B on the different BC algorithms and their specific loss functions and also note that 4.4.1 and 4.4.2 provide details on the implicit attacks. Please let us know if this addresses your concern.
>
>
>
> > Add experiments of applying robust RL attack instead of supervised learning attack, which makes the paper much more reasonable.
>
> We thank the reviewer for pointing out "Robust Deep Reinforcement Learning against Adversarial Perturbations on State Observations" (Zhang et al., NeurIPS 2020). We note that the paper proposes two attacks on RL policies. The first attack, Robust Sarsa (RS) Attack, seeks to identify adversarial perturbations that reduce the value of an agent’s policy. They do this by training a new critic from on-policy rollouts using Sarsa-style TD learning and regularize this critic to be robust and then perform an iterative gradient-based attack (similar to PGD) to find a perturbed state that causes the agent to take an action with lower value at the original state.  We note that this is specifically designed for reinforcement learning (RL) settings where policies are trained to maximize expected reward, and adversarial perturbations are evaluated or optimized based on their impact on return. In contrast, our work is situated entirely in the imitation learning (IL) setting, where no reward signal is available, and policies are learned purely via supervised behavior cloning from demonstrations. As such, the core mechanisms in these RL methods—including adversarial value estimation, robust policy gradients, and environment interactions for adversarial training—do not apply in our setting.
> We note that Zhang et al. (2020) also propose a Maximal Action Difference (MAD) Attack which does not consider the reward, but seeks to maximize the action change caused by a perturbation. We note that this setting is similar to our undirected attack setting, but that we also consider targeted attacks and universal attacks which are not considered in prior work.
>
>
> We agree that the follow-on work by Zhang et al. (2021) proposes an even more sophisticated state-based attack; however, the new attack mechanism (ATLA) also relies on the RL setting where a reward is available and where on-policy rollouts and RL can be performed. The adversary is parameterized as a policy network trained using standard RL with the goal of choosing perturbations such that the agent being attacked receives as little reward as possible. Thus, the adversary tries to maximize the negative of the agent’s reward function. We agree that this is an interesting attack, but it is out of scope given our focus on offline attacks (no access to the environment) and imitation learning (no access to the reward function).
>
> We thank you for the additional robust RL citations and have added them to our paper.
>
>
> >Add more experiments results from other domains, e.g., D4RL benchmark.
>
> Thank you for your suggestion. We note that the standard benchmarks often used from D4RL use low-dimensional state information rather than images. We are in the process of adding a longer horizon image kitchen task based on the Franka Kitchen task from D4RL. We use the Franka Kitchen robotic manipulation environment introduced in (Gupta et al., 2019) with a Franka Panda arm with a 7 dimensional action space. This environment requires the agent to complete a sequence of four tasks in order. While the original environment is state-based, we use an image-based variant of it by rendering the states with the MuJoCo renderer as a 112 by 112 image. We currently have results for VQ-BET showing that PGD attacks on VQBET drops our success rate (expected number of tasks in the 4 task sequence that are completed) from 2.86 to 0. Given this promising result on a more complex image-based adaptation of a D4RL benchmark we plan to add results for BC, LSTM-GMM, IBC, and Diffusion Policy.
>
>
> > Since the title is "How vulnerable is my policy" and the key objective of this pape is to study the vulnerability of the BC policies, I believe the authors should put the fig.11 (the sensitivity to adversarial range) to main text instead of appendix, which provides the key information for how vulnerable the policy is.
>
> Great point. We agree that this is an important result that strengthens our paper and have moved these results to the main paper in section 5.8 and included this as one of our experimental questions at the beginning of Section 5.

---

### Review · Reviewer_J1z3 · 2025-04-09

**Summary Of Contributions:**

# Summary

This paper presents a comprehensive study of adversarial attacks on various state-of-the-art behavior cloning (BC) algorithms used in robotic manipulation. The authors analyze both explicit (e.g., Vanilla BC, LSTM-GMM, VQ-BET) and implicit (e.g., IBC, Diffusion Policy) policy learning methods under white-box attack scenarios. They propose tailored attack strategies for complex policy architectures—particularly for implicit models like IBC and diffusion-based policies—and evaluate the impact of both targeted and untargeted perturbations across several environments and backbones.

Their findings reveal that:
- All tested policies are vulnerable to even minor adversarial perturbations.
- Implicit models (e.g., Diffusion Policy, IBC) demonstrate relatively better robustness than explicit ones.
- Perturbations transfer well across models, tasks, and backbones.
- Randomized smoothing as a defense mechanism shows limited success, especially in complex, multimodal environments.

**Audience:**

Yes

**Claims And Evidence:**

Yes

**Requested Changes:**

# Required Changes

1. **Clarify Evaluation Metrics and Improve Visualizations**:
   The authors should more clearly define the meaning of metrics in each table—specifically whether they are showing success rates or success rate reductions. One way to improve clarity is to **standardize the metric format across all tables** and clearly indicate the unit (e.g., percentage of task success or percentage drop). Additionally, consider using **bar graphs like those in Figures 2 and 3** to visualize results from tables such as Table 3. This would make anomalies—such as the counterintuitive “improvement” seen in BC's transfer result—easier to interpret. For instance, if the absolute success rate is already low, a 40% improvement may not be practically meaningful.

2. **Deeper Analysis on IBC vs. Diffusion Policy Robustness**:
   The observed difference in robustness patterns between IBC and Diffusion Policy deserves further explanation. For example, is Diffusion Policy more robust to PGD attacks due to its multi-step inference mechanism that dilutes the effect of small perturbations? On the other hand, does IBC's sampling-based action selection naturally resist universal perturbations due to energy surface diversity? A deeper theoretical or empirical analysis—possibly through ablation or step-by-step tracing—would improve understanding.

3. **Optional but Encouraged: Include an Additional Implicit Baseline**:
   If computational resources permit, the authors might consider including another implicit policy baseline, such as ACT (Zhao et al.) (which uses a VAE-style latent space for behavior modeling). Comparing it with IBC and DP could help disentangle whether the observed robustness stems from stochasticity, multi-step inference, or other architectural properties. This could also provide a more comprehensive view of the design space for robust LfD models.

Zhao, T. Z., Kumar, V., Levine, S., & Finn, C. (2023). Learning fine-grained bimanual manipulation with low-cost hardware. arXiv preprint arXiv:2304.13705.

**Strengths And Weaknesses:**

# Strengths

1. **Thorough Empirical Evaluation**: The paper covers five prominent BC methods across four manipulation tasks, using both PGD and UAP attacks. It explores not only direct attack success but also transferability across tasks, architectures, and models.

2. **Novel Attack Strategies for Implicit Policies**: The proposed attack formulations for IBC and Diffusion Policy are thoughtful and appropriately address the unique characteristics of these models.

3. **Clear and Well-Structured**: The paper is easy to follow, with a clear threat model, problem formulation, and well-structured experiments. Algorithms are described in detail, and figures/tables are used effectively.

4. **Relevant and Timely Topic**: As BC methods are increasingly being used in real-world robotics, understanding their robustness is critical. The topic is both practically important and underexplored.

5. **Insightful Findings**: For example, the observation that attacks on later steps of the diffusion process are more effective and computationally efficient is a useful insight for future robustness research.

# Weaknesses

1. **Evaluation Metrics Clarity**:
   While the use of task success rate is common in robotics, the paper does not consistently clarify what each metric in the tables represents. For instance, in some tables (e.g., Table 3), it's unclear whether the values denote absolute success rates or relative performance drop percentages. The mix of percentage formats (e.g., "100%") and decimal formats (e.g., "0.65") without explicit clarification may confuse readers. Although close reading allows readers to infer the meaning, clearer and more consistent metric labeling would significantly improve readability.

2. **Insufficient Analysis on Robustness Differences Between IBC and Diffusion Policy**:
   While the paper points out that implicit policies like IBC and Diffusion Policy are generally more robust than explicit ones, the underlying reasons are not thoroughly analyzed. The explanation that their robustness comes from “inherent stochasticity” is too high-level and not fully supported by experimental dissection. For instance, Figure 2 and 3 suggest that **IBC is more robust to UAP**, whereas **Diffusion Policy is more robust to PGD**, yet the paper does not explain this asymmetry. A deeper analysis of why different attack types affect these models differently would add valuable insight.

---

> ### Author Response · Authors · 2025-04-24
>
> Thank you for your comments and helpful suggestions and questions! We thank you for noticing that our paper has a thorough empirical evaluation, novel attack strategies, is clear and well-structured, and covers a relevant and timely topic with insightful findings
>
>
> > The authors should more clearly define the meaning of metrics in each table—specifically whether they are showing success rates or success rate reductions.
>
> Thank you for this suggestion. We have revised our paper, making sure to define the meaning of metrics in each table. We agree that this will help readability.
>
>
> > Deeper Analysis on IBC vs. Diffusion Policy Robustness:
>
> Thank you for pointing this out. We agree that the robustness patterns between IBC and Diffusion Policy are interesting. We have added a discussion section to Section 5.5. We also note that Section 5.7 already performs some analysis of the action prediction horizon of Diffusion Policy. We note that fully understanding the empirical differences between the vulnerability of Diffusion Policy and IBC is difficult, but agree that further empirical and theoretical understanding of these phenomena is an interesting area for future work. We believe that our initial findings will lead to further interesting research along these directions and do not think it is a significant weakness that we don’t yet have an explanation for all of our empirical findings.
>
> > If computational resources permit, the authors might consider including another implicit policy baseline, such as ACT (Zhao et al.) (which uses a VAE-style latent space for behavior modeling).
>
> Thank you for the suggestion. We would like to note that VQ-BeT uses a variant of a VAE—specifically a Residual Vector-Quantized VAE as part of its architecture. Furthermore, VQ-BET was proposed after ACT. Thus, we believe that VQ-BET already provides a good comparison with a VAE-based and Transformer-based modern BC method that is more recent than ACT.

---

### Review · Reviewer_Ho1u · 2025-04-09

**Summary Of Contributions:**

this paper presents a study for adversarial attacks on several classic and recently proposed behavior cloning algorithms, including behavior cloning, LSTM-GMM, IBC, Diffusion Policy and VQ-BET. They show that explicit policies such as BC, LSTM-GMM and VQ-BET can be attacked in the same manner as standard computer vision models, while showed methods for attacking implicit policies such as Diffusion Policy and IBC. The study also show some transferability across algorithms, architectures and tasks regarding the adversarial attack.

**Audience:**

No

**Broader Impact Concerns:**

this work does not have concerns on ethical implications.

**Claims And Evidence:**

No

**Requested Changes:**

instead of saying the contributions include attacking classical methods like BC, LSTM-GMM, only claiming the part that are truly innovating, and provide clear formulation of the algorithm, including the reason why the equations are defined in the way that is presented in the paper.

It would also be more desirable to check if such attacks can be applied to non-vision based scenarios or scenarios that involve a hybrid observation space including force sensing and vision. Since in scenarios where vision and force sensing are used as feedback, adversarial attack on vision may not be that significant to defeat a policy.

**Strengths And Weaknesses:**

The study shows the adversarial robustness of several classic and recent BC methods on some simulated robotics manipulation tasks. Especially it shows a more computationally efficient attack methods on diffusion policy that does not involve attacking the entire denoising process. They also examines the vulnerability when attacks transfer across different algorithms, tasks and visual backbones. The main novelty is in the proposal of attacks on implicit behavior cloning and diffusion policy.

The main weakness is in the motivation of the work, and technical novelty. Since adversarial attacks on BC models such as BC, LSTM-GMM, has been explored before in various related works, it is unclear why the study includes those as part of their contributions. Besides, the algorithms are proposed without appropriate mathematical formulations and derivations. Another weakness is that the formulation still considers the victim model to be a white-box accessible model, which in many cases aren't true.

---

> ### Author Response · Authors · 2025-04-24
>
> Thank you for your comments and helpful suggestions and questions!
>
> >The main weakness is in the motivation of the work, and technical novelty. Since adversarial attacks on BC models such as BC, LSTM-GMM, has been explored before in various related works, it is unclear why the study includes those as part of their contributions.
>
> As we note in the intro, we are the first to present a systematic study of the vulnerabilities of LfD algorithms to adversarial attacks with a focus on modern behavior cloning algorithms. We have clarified this in the contributions section as well as rearranged our paper to have a dedicated related work section to make our contributions relative to prior work clear.
>
> Based on a thorough literature search we have not found any other prior work that performs attacks on LSTM-GMM, VQ-BET, or IBC. We did find a few papers that study adversarial attacks on vanilla BC policies and have added references to them and discussions in the related work section. If there are other papers that study adversarial attacks on BC or LSTM-GMM that you are aware of, please let us know.
>
>
> >The algorithms are proposed without appropriate mathematical formulations and derivations.
>
> Based on your suggestion, we have added more mathematical details to Appendix B for the different algorithms. We agree that these details are important and having them explicitly stated in the Appendix will strengthen the paper. We have added a reference to the relevant appendix section to our intro for Section 3.
>
>
> >Another weakness is that the formulation still considers the victim model to be a white-box accessible model
>
> Thank you for bringing this up. We would like to emphasize that our transfer attacks are actually a form of black-box attack, where a white-box attack is transferred to policy with unknown architecture and weights. We believe this point is important to make clear and have added emphasis of our black-box attack analysis throughout the paper. We agree that this kind of analysis is important since transfer attacks are common and practical methods for performing blackbox attacks.
>
> > Instead of saying the contributions include attacking classical methods like BC, LSTM-GMM, only claiming the part that are truly innovating,
>
> Please see our response above. We have clarified our claims of novelty. To the best of our knowledge, we are the first to study and compare attacks across different BC models, the first to study any kind of attack on LSTM-GMM, IBC, and VQ-BET, the first to study a range of attacks (targeted, untargeted, and universal), and the first to study black-box transfer attacks across algorithms, architectures, and tasks.
>
> >Provide clear formulation of the algorithm, including the reason why the equations are defined in the way that is presented in the paper.
>
> As requested, we have added details to Appendix B. We also note that we already do include a detailed discussion of the novel implicit attacks in the main paper in sections 4.4.1 and 4.4.2. We believe our notation is not unusual, but please let us know if there are equations or definitions that you feel are confusing and we would be happy to try and edit them to make things more readable.
>
> >It would also be more desirable to check if such attacks can be applied to non-vision based scenarios or scenarios that involve a hybrid observation space including force sensing and vision. Since in scenarios where vision and force sensing are used as feedback, adversarial attack on vision may not be that significant to defeat a policy.
>
> We agree that hybrid attacks are interesting since many modern BC methods use a combination of both pixel and vector-based state inputs. We would like to clarify that we do already study these kinds of hybrid attacks. In Robomimic, the policy uses state information that includes the proprioception, including the eef pose and eef quaterions and gripper state as input along with the camera images. Our attacks are only applied to the vision component of the state. We agree that this is an important aspect and have added this clarification to Section 4.2 to point this out. Our current results show that even in settings where the robot feedback is more than just vision, adversarial attacks only on vision are significant enough to defeat a well-trained imitation policy. We agree that investigating domains with force sensing feedback is an exciting area for future works and we have added this to the conclusion.

---

### Decision · Action_Editor_wei4 · 2025-05-26

**Recommendation:** Reject

**Comment:**

We thank the authors for their submission. While the paper presents an interesting direction with meaningful empirical observations, especially concerning robustness in policy learning, the reviewers collectively raised important concerns regarding the paper’s novelty, depth of analysis, and experimental rigor. In particular,  limited theoretical insight into the observed robustness differences and the lack of real-world robotic experiments were viewed as significant weaknesses, especially given the focus of the paper.

Given these limitations, we are unable to recommend acceptance in its current form. That said, we appreciate the potential of this line of work and encourage the authors to further develop the theoretical grounding and experimental evaluation, and consider resubmitting a revised version in the future.

Regards,
AC

**Audience:**

Yes

**Claims And Evidence:**

No

**Resubmission Of Major Revision:**

The authors may consider submitting a major revision at a later time.